# Homeostatic interferon-lambda response to bacterial microbiota stimulates preemptive antiviral defense within discrete pockets of intestinal epithelium

Jacob A Van Winkle[1], Stefan T Peterson[2], Elizabeth A Kennedy[2], Michael J Wheadon[2], Harshad Ingle[2], Chandni Desai[3], Rachel Rodgers[2], David A Constant[1], Austin P Wright[1], Lena Li[1], Maxim N Artyomov[4], Sanghyun Lee[2], Megan T Baldridge[5]*, Timothy J Nice[1]*

[1]Department of Molecular Microbiology and Immunology, Oregon Health & Science University, Portland, United States; [2]Department of Medicine, Washington University School of Medicine, St. Louis, United States; [3]Department of Pathology and Immunology, Washington University School of Medicine, St. Louis, United States; [4]Department of Pathology and Immunology, Washington University School of Medicine, St Louis, United States; [5]Department of Medicine, Washington University School of Medicine, St Louis, United States

*For correspondence:
mbaldridge@wustl.edu (MTB);
nice@ohsu.edu (TJN)

**Competing interest:** The authors declare that no competing interests exist.

**Abstract** Interferon-lambda (IFN-$\lambda$) protects intestinal epithelial cells (IECs) from enteric viruses by inducing expression of antiviral IFN-stimulated genes (ISGs). Here, we find that bacterial microbiota stimulate a homeostatic ISG signature in the intestine of specific pathogen-free mice. This homeostatic ISG expression is restricted to IECs, depends on IEC-intrinsic expression of IFN-$\lambda$ receptor (*Ifnlr1*), and is associated with IFN-$\lambda$ production by leukocytes. Strikingly, imaging of these homeostatic ISGs reveals localization to pockets of the epithelium and concentration in mature IECs. Correspondingly, a minority of mature IECs express these ISGs in public single-cell RNA sequencing datasets from mice and humans. Furthermore, we assessed the ability of orally administered bacterial components to restore localized ISGs in mice lacking bacterial microbiota. Lastly, we find that IECs lacking *Ifnlr1* are hyper-susceptible to initiation of murine rotavirus infection. These observations indicate that bacterial microbiota stimulate ISGs in localized regions of the intestinal epithelium at homeostasis, thereby preemptively activating antiviral defenses in vulnerable IECs to improve host defense against enteric viruses.

## Editor's evaluation

The paper shows that homeostatic interferon-stimulated gene expression in the mouse intestinal epithelium (which is not uniform but concentrated in mature epithelial pockets) depends on the presence of bacterial microbiota and intestinal epithelial cell-intrinsic expression of the IFN-$\lambda$ receptor associated with leucocyte IFN-$\lambda$ production. Mouse rotavirus infection (an intestinal epithelial pathogen) is more effectively initiated in the absence of this homeostatic IFN-$\lambda$ response although it remains unclear how the localized pockets of interferon-stimulated gene expression impart protection.

## Introduction

Interferons (IFNs) are a family of cytokines produced in response to infection that signal IFN receptor-bearing cells to induce transcription of hundreds of IFN-stimulated genes (ISGs). These ISGs perform diverse functions, but many cooperate to induce an antiviral state (*Sadler and Williams, 2008*; *Schoggins and Rice, 2011*). There are three types of IFNs: type I IFNs (IFN-αs, IFN-β, others), type II IFN (IFN-γ), and type III IFNs (IFN-$\lambda$s). These three types are differentiated by receptor usage (type I IFN receptor: *Ifnar1/Ifnar2*; type II IFN receptor: *Ifngr1/Ifngr2*; type III IFN receptor: *Ifnlr1/Il10rb*), but all three receptor complexes signal through Janus-kinase (JAK) and signal transducer and activator of transcription (STAT) factors to stimulate ISG transcription (*Ingle et al., 2018*; *Schneider et al., 2014*). Type I and III IFNs are directly stimulated by host detection of microbe-associated molecular patterns (MAMPs) such as viral nucleic acids, and the prominent contribution of these IFN types to antiviral defense is reflected by the breadth of evasion strategies used by diverse viral families to prevent their production or action (*Levy and García-Sastre, 2001*; *Rojas et al., 2021*; *Taylor and Mossman, 2013*). The type I IFN receptor is expressed broadly across most cell types, whereas the type III IFN receptor, *Ifnlr1,* is primarily restricted to epithelial cells (*Kotenko and Durbin, 2017*; *Sommereyns et al., 2008*). Accordingly, IFN-$\lambda$ is of particular importance in effective antiviral defense of barrier tissues.

Interestingly, previous studies in mice have noted that intestinal epithelial cells (IECs) are hyporesponsive to type I IFN (*Lin et al., 2016*; *Hernández et al., 2015*; *Pott et al., 2011*; *Van Winkle et al., 2020*). The responsiveness of IECs to type I IFN appears to be developmentally regulated because the IECs of adult mice exhibit weaker type I responses than IECs of neonatal mice (*Lin et al., 2016*). Additionally, infection of mice deficient in IFN receptors (*Ifnar1, Ifngr1, Ifnlr1*; single or double knockouts) with enteric viruses indicates that IFN-$\lambda$ is the predominant antiviral IFN type that controls viral replication in the gastrointestinal epithelium (*Hernández et al., 2015*; *Nice et al., 2015*; *Pott et al., 2011*). IECs can robustly respond to IFN-$\lambda$ with upregulation of canonical antiviral ISGs and increased resistance to infection by enteric viruses, such as rotaviruses and noroviruses (*Baldridge et al., 2017*; *Nice et al., 2015*; *Pott et al., 2011*). Mouse rotavirus is a natural pathogen of mice that infects enterocytes located at the tips of villi in the small intestine (*Burns et al., 1995*). Rotaviruses have developed mechanisms to antagonize the induction of IFN by infected cells, suggesting that evasion of the IFN response is necessary for efficient epithelial infection (*Arnold et al., 2013*). Indeed, prophylactic administration of IFN-$\lambda$ significantly reduces the burden of mRV infection, demonstrating its potential for mediating epithelial antiviral immunity to this pathogen (*Mahlakõiv et al., 2015*; *Lin et al., 2016*; *Pott et al., 2011*; *Van Winkle et al., 2020*). However, a protective role of the endogenous IFN-$\lambda$ response to mRV infection is less clear, perhaps reflecting the success of mRV evasion mechanisms.

Epithelial immunity in the gut must be appropriately balanced to protect the intestinal epithelium while preventing loss of barrier integrity and intrusion by microbes that are abundant within the gastrointestinal tract. The bacterial microbiota in the intestine perform critical functions by aiding in host metabolism (*Krajmalnik-Brown et al., 2012*), providing a competitive environment to defend against pathogens (*Kim et al., 2017*), and initiating and maintaining host immune function during homeostasis (*Honda and Littman, 2016*; *Rooks and Garrett, 2016*). In this complex environment host epithelial and immune cells detect bacteria and viruses using a suite of pattern recognition receptors (PRRs) that sense the presence of MAMPs. Stimulation of PRRs, such as the toll-like receptor (TLR) family, activates antimicrobial and antiviral defenses providing local protection in many tissues (*Chu and Mazmanian, 2013*; *Iwasaki and Medzhitov, 2015*; *Thompson et al., 2011*). TLR-dependent pathways induce production of IFNs, primarily by signaling through TIR-domain-containing adapter-inducing interferon-β (TRIF) and myeloid differentiation primary response 88 (MYD88) adapter proteins (*Monroe et al., 2010*; *Odendall et al., 2017*), providing a mechanism by which bacterial MAMPs can initiate IFN responses.

Signals from the bacterial microbiota have been shown to elicit a steady-state type I IFN response in systemic tissues and cell types that can prime antiviral immunity by several independent studies (*Abt et al., 2012*; *Bradley et al., 2019*; *Ganal et al., 2012*; *Steed et al., 2017*; *Stefan et al., 2020*; *Winkler et al., 2020*). Additionally, a steady-state ISG signal has been observed in the intestine of uninfected mice (*Baldridge et al., 2015*; *Lin et al., 2016*; *Stockinger et al., 2014*), but this intestinal response remained poorly characterized. Together with the observed hypo-responsiveness of IECs to type I IFN, these findings suggested that bacterial microbiota may stimulate an IFN-$\lambda$ response in the gut. To explore this interaction, we undertook the present study to assess the role of bacterial

microbiota in induction of enteric ISGs at homeostasis using a combination of broad-spectrum antibiotics (ABX) and genetically modified mice.

In this study, we uncovered an ISG signature in IECs that was dependent on the presence of bacteria and IFN-$\lambda$ signaling (hereafter referred to as 'homeostatic ISGs'). This panel of genes was present in wild-type (WT) mice with conventional microbiota and was reduced in WT mice treated with ABX and in mice lacking *Ifnlr1*. We revealed that homeostatic ISG expression is (i) restricted to the intestinal epithelium across both the ileum and colon, (ii) independent of type I IFN signaling, and (iii) associated with IFN-$\lambda$ transcript expression by epithelium-associated CD45+ leukocytes. Surprisingly, we found that homeostatic ISGs are not expressed uniformly by all IECs; rather, expression is concentrated in localized pockets of IECs and in differentiated IECs relative to crypt-resident progenitors. These patterns of localized ISG expression are corroborated by independently generated single-cell RNA sequencing (scRNA-seq) data from mouse and human IECs. We also found that ISG expression can be increased in ABX-treated mice by reconstitution of bacterial microbiota or administration of bacterial lipopolysaccharide (LPS). Finally, we found that this microbiota-stimulated ISG signature provides protection from initiation of mRV infection. Cumulatively, this study found that bacteria initiate preemptive IFN-$\lambda$ signaling in localized areas to protect IECs from enteric viruses.

## Results
### Bacterial microbiota stimulate IFN-$\lambda$ response genes in the ileum at homeostasis

To determine the effect of bacterial microbiota on homeostatic IFN-$\lambda$ responses, we compared gene expression in whole ileum tissue for the following experimental groups: (i) wild-type (WT) C57BL/6J mice intraperitoneally injected with IFN-$\lambda$ as compared to unstimulated WT mice, (ii) WT mice with conventional microbiota as compared to WT mice treated with an antibiotic cocktail (ABX) to deplete bacteria, (iii) WT mice with conventional microbiota as compared to *Ifnlr1*-/- mice with conventional microbiota, and (iv) *Ifnlr1*-/- mice with conventional microbiota compared to *Ifnlr1*-/- mice treated with ABX (*Figure 1A*). To rule out contributions of *Ifnlr1* toward an altered intestinal bacterial microbiota, we performed 16S rRNA sequencing on stool from *Ifnlr1*+/+, *Ifnlr1*+/-, and *Ifnlr1*-/- mice and did not find statistically significant differences in alpha-diversity and beta-diversity measurements with the statistical power available from 12 mice per group (*Figure 1—figure supplement 1*). For each comparison of ileum gene expression shown in *Figure 1A*, we performed gene set enrichment analysis (GSEA) to determine enrichment or depletion of hallmark gene sets (*Liberzon et al., 2015*). The hallmark gene set that was most enriched in IFN-$\lambda$-treated WT mice compared to unstimulated WT mice was INTERFERON_ALPHA_RESPONSE (*Figure 1B*, *Figure 1—figure supplement 2*). Therefore, this gene set reflects differences in IFN-$\lambda$ responses between experimental conditions.

To deplete bacterial microbiota, we administered a cocktail of broad-spectrum ABX, and demonstrated that this treatment reduced bacterial 16S gene copies in stool to below the limit of detection (*Figure 1—figure supplement 3*). GSEA of WT mice treated with ABX showed a significant depletion (negative enrichment score) of INTERFERON_ALPHA_RESPONSE hallmark genes in the ileum relative to WT mice with conventional microbiota (*Figure 1C*). These data indicate that ISGs are present at steady state in the ileum of specific-pathogen-free mice with conventional microbiota and suggest that microbiota stimulate expression of these genes in the ileum at homeostasis.

Type I, II, and III IFN responses have substantial gene expression overlap; therefore, to prove that the IFN-$\lambda$ receptor was necessary for expression of ISGs at homeostasis, we analyzed gene expression in the ileum of *Ifnlr1*-/- mice. Indeed, GSEA showed significant reduction of INTERFERON_ALPHA_RESPONSE hallmark genes in the ileum of *Ifnlr1*-/- mice compared to WT mice (*Figure 1D*). Furthermore, expression of INTERFERON_ALPHA_RESPONSE hallmark genes was not significantly decreased in *Ifnlr1*-/- mice treated with ABX compared to *Ifnlr1*-/- mice with conventional microbiota (*Figure 1E*). In contrast to INTERFERON_ALPHA_RESPONSE, other hallmark gene sets such as INFLAMMATORY_RESPONSE and IL6_JAK_STAT3_SIGNALING were significantly decreased by ABX treatment in both WT mice and *Ifnlr1*-/- mice (*Figure 1—figure supplement 2*), which indicates a selective requirement for *Ifnlr1* to elicit ISGs and a minimal effect of *Ifnlr1* deficiency on other microbiota-dependent genes. Together, these findings suggest that *Ifnlr1* is necessary for the bacterial microbiota-dependent expression of ISGs in the ileum at homeostasis. Since ISGs are enriched in

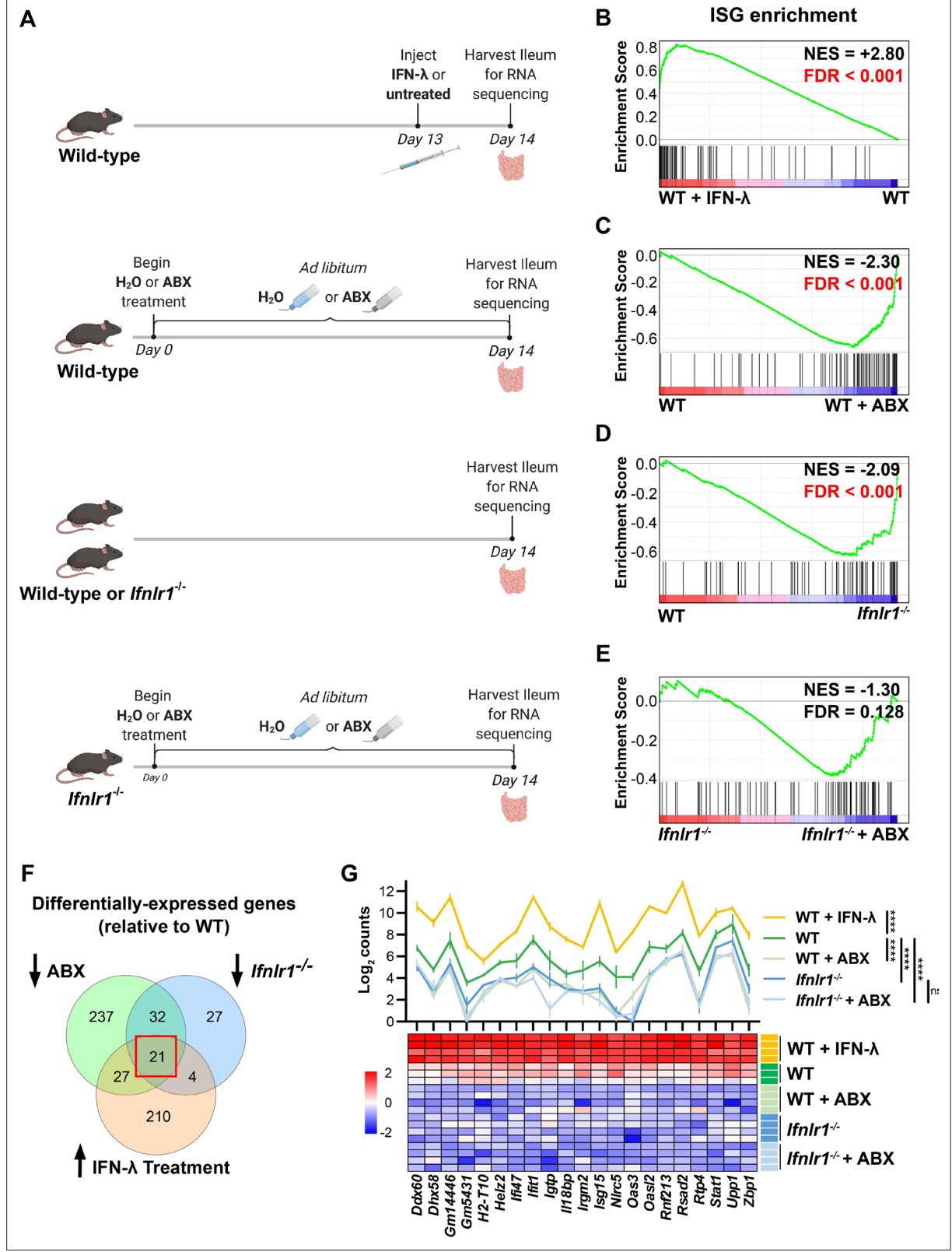

**Figure 1.** Bacterial microbiota stimulate interferon-lambda (IFN-λ) response genes in the ileum at homeostasis.
(**A**) Depiction of experimental treatments and comparison groups. Following the indicated treatments, a segment of whole ileum tissue was harvested and analyzed by RNA sequencing for differentially expressed genes between paired conditions. (**B–E**) Gene set enrichment analysis of INTERFERON_ALPHA_RESPONSE hallmark genes (interferon-stimulated genes [ISGs]) was performed with the following comparisons: wild-type (WT) mice treated

*Figure 1 continued on next page*

*Figure 1 continued*

with 25 µg of IFN-$\lambda$ relative to WT mice treated with PBS (**B**), WT mice treated with antibiotics (ABX) relative to untreated WT mice (**C**), *Ifnlr1*$^{-/-}$ mice relative to WT mice (**D**), and *Ifnlr1*$^{-/-}$ mice treated with ABX relative to untreated *Ifnlr1*$^{-/-}$ mice (**E**). Normalized enrichment score (NES) and false discovery rate (FDR) are overlaid for each comparison with significant FDR's highlighted (red). (**F**) A Venn diagram depicting the total number of differentially expressed genes that are (i) increased with IFN-$\lambda$ stimulation (orange), (ii) decreased with ABX treatment (green), or (iii) decreased in *Ifnlr1*$^{-/-}$ mice relative to WT (blue). An overlapping subset of 21 genes was shared among all three comparisons (red box). (**G**) A graph and heatmap of the relative expression of the 21 genes that overlap in all experimental groups in (**F**) ('homeostatic ISGs'). Statistical significance in (**G**) was determined by one-way ANOVA with Tukey's multiple comparisons where ns = p > 0.05 and **** = p < 0.0001.

The online version of this article includes the following source data and figure supplement(s) for figure 1:

**Source data 1.** Gene set enrichment analysis, differential gene expression analysis, and counts of homeostatic interferon-stimulated genes (ISGs) from RNA sequencing analysis.

**Figure supplement 1.** *Ifnlr1* deficiency does not alter intestinal bacterial microbiota.

**Figure supplement 1—source data 1.** Values and statistical tests of Observed Species and Shannon Diversity Index from 16S rRNA sequencing.

**Figure supplement 2.** Gene set enrichment analysis of hallmark gene sets.

**Figure supplement 2—source data 1.** Full results of gene set enrichment analysis for HALLMARK genes in *Figure 1* RNA sequencing dataset.

**Figure supplement 3.** Treatment with antibiotics (ABX) reduces enteric 16S gene copies to below the limit of detection.

**Figure supplement 3—source data 1.** Values and statistical test of 16S gene copies with antibiotics (ABX) treatment.

mice upon IFN-$\lambda$ stimulation and are decreased in *Ifnlr1*$^{-/-}$ and ABX-treated mice, we conclude that the bacterial microbiota stimulates IFN-$\lambda$ responses in the ileum at homeostasis.

To define a core set of bacterial microbiota-dependent, *Ifnlr1*-dependent 'homeostatic ISGs', we determined the overlap of differentially expressed genes (DEGs) by defining genes with: (i) increased expression upon IFN-$\lambda$ stimulation in WT mice, (ii) decreased expression upon ABX treatment in WT mice, and (iii) decreased expression in *Ifnlr1*$^{-/-}$ mice relative to WT mice. The DEGs shared by each of these comparisons comprised a set of 21 genes that are decreased upon treatment with ABX and loss of *Ifnlr1*, and are induced in response to IFN-$\lambda$ (*Figure 1F*). This set of homeostatic ISGs includes antiviral genes that are dependent on bacterial microbiota and the IFN-$\lambda$ pathway. Comparison of homeostatic ISG transcript counts between experimental treatments revealed similar insights as prior GSEA. WT mice treated with IFN-$\lambda$ had higher expression of all homeostatic ISGs than untreated mice, whereas WT mice with a conventional microbiota had higher expression of homeostatic ISGs than *Ifnlr1*$^{-/-}$ mice and ABX-treated mice of both genotypes (*Figure 1G*). We did not detect additional decreases in these homeostatic ISGs in ABX-treated *Ifnlr1*$^{-/-}$ mice relative to conventional *Ifnlr1*$^{-/-}$ mice, suggesting that *Ifnlr1* is necessary for expression of homeostatic ISGs (*Figure 1G*). These results indicate that there is modest but significant expression of ISGs at homeostasis that is lost with *Ifnlr1* deficiency or ABX treatment. Together, these analyses revealed a homeostatic signature of ISGs in the ileum that depends upon the presence of bacterial microbiota and on intact IFN-$\lambda$ signaling.

## Homeostatic, microbiota- and *Ifnlr1*-dependent ISGs are primarily expressed in intestinal tissues

To complement the results of the RNA-seq and extend this analysis to other tissue sites, we quantified tissue-level expression of a panel of three ISGs by quantitative PCR (qPCR): *Ifit1*, *Oas1a*, and *Stat1*. *Ifit1* and *Stat1* were present among the 21 homeostatic ISGs in the preceding analysis and *Oas1a* was included as a representative canonical ISG that we hypothesized would be present in the homeostatic signature, but did not meet the statistical criteria used to define the core set of 21 homeostatic ISGs (*Figure 1F–G*). We assessed absolute abundance of these ISG transcripts in the ileum, colon, mesenteric lymph nodes (MLN), and spleen tissue of WT and *Ifnlr1*$^{-/-}$ mice with or without ABX treatment (*Figure 2A–C*). Consistent with our RNA-seq data, these ISGs were reduced in the ilea of *Ifnlr1*$^{-/-}$ mice and ABX-treated WT mice compared to WT mice with conventional microbiota (*Figure 2A*). Second, homeostatic ISGs were expressed in WT colonic tissue and were significantly decreased in colonic tissue of *Ifnlr1*$^{-/-}$ mice and ABX-treated WT mice (*Figure 2B*). These data indicate that homeostatic ISGs in both the ileum and colon were dependent on *Ifnlr1* and the bacterial microbiota; however, these homeostatic ISGs were more abundantly expressed in ileal tissue than colonic tissue (*Figure 2—figure supplement 1*). To confirm that treatment with ABX does not ablate the ability of the intestine to respond to IFN-$\lambda$, we stimulated ABX-treated mice with intraperitoneal

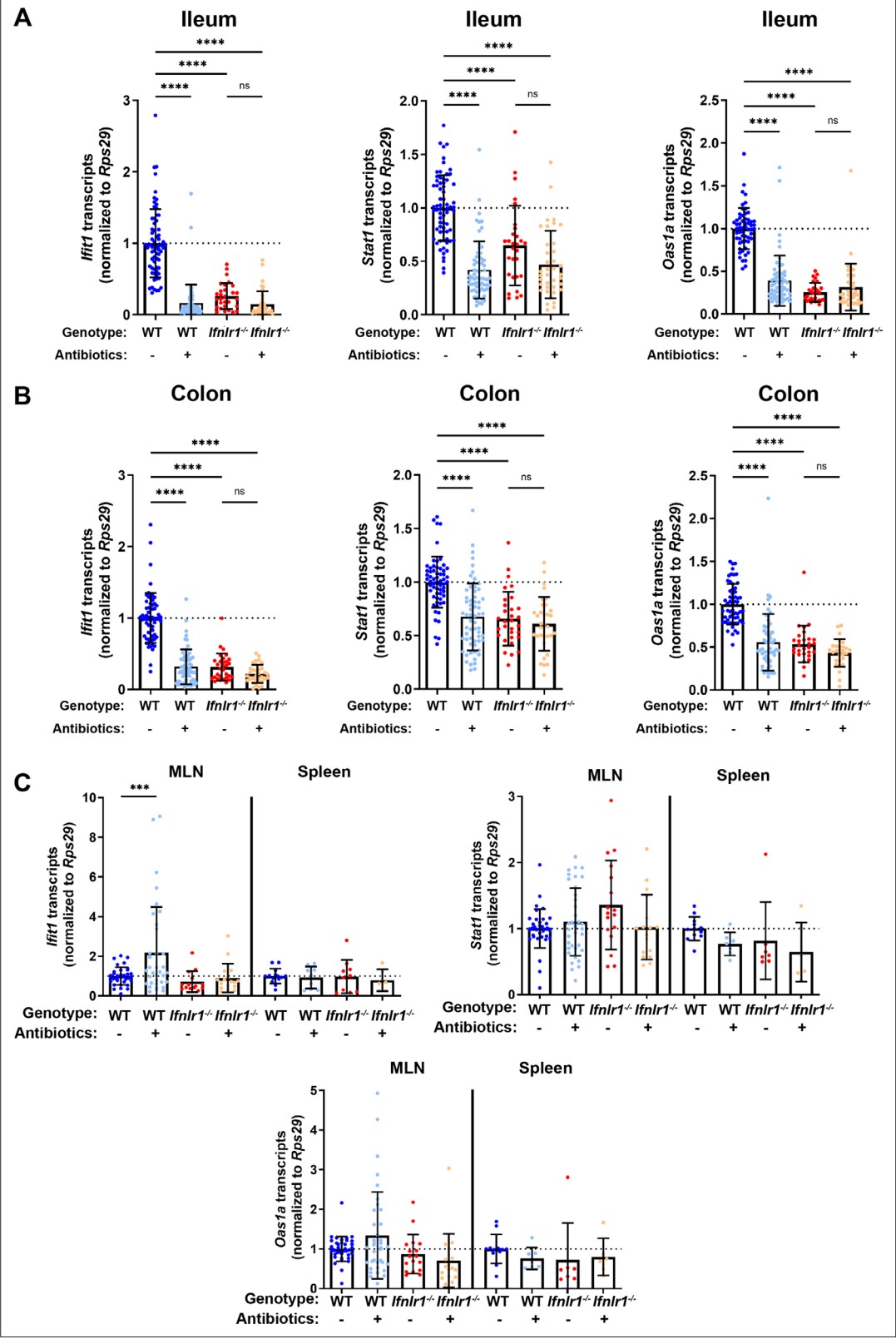

**Figure 2.** Homeostatic, microbiota- and *Ifnlr1*-dependent interferon-stimulated genes (ISGs) are primarily expressed in intestinal tissues. A segment of ileum or colon tissue from wild-type (WT) or *Ifnlr1⁻/⁻* mice was harvested following H$_2$O or antibiotics (ABX) treatment and the ISGs *Ifit1*, *Stat1*, and *Oas1a* were analyzed by quantitative PCR (qPCR). Transcripts were quantified in ileum (**A**), colon (**B**), or mesenteric lymph node (MLN) and

*Figure 2 continued on next page*

*Figure 2 continued*

spleen (**C**) with normalization to untreated WT mice. Data points represent individual mice and data are pooled from 10 to 20 independent experiments in (**A–B**) and 15 independent experiments in (**C**). Statistical significance was determined by one-way ANOVA with Tukey's multiple comparisons in (**A–B**) and Dunnett's multiple comparisons in (**C**). * = $p < 0.05$, ** = $p < 0.01$, *** = $p < 0.001$, **** = $p < 0.0001$.

The online version of this article includes the following source data and figure supplement(s) for figure 2:

**Source data 1.** Values and statistical tests of interferon-stimulated genes (ISG) expression in ileum, colon, mesenteric lymph node (MLN), and spleen.

**Figure supplement 1.** Homeostatic interferon-stimulated genes (ISGs) are more abundantly expressed in the ileum than in the colon.

**Figure supplement 1—source data 1.** Values and statistical tests of interferon-stimulated gene (ISG) expression between the ileum and colon.

**Figure supplement 2.** Treatment with antibiotics (ABX) does not ablate responsiveness to interferon-lambda (IFN-$\lambda$).

**Figure supplement 2—source data 1.** Values and statistical tests of interferon-stimulated gene (ISG) expression in antibiotics (ABX)-treated mice stimulated with interferon-lambda (IFN-$\lambda$).

IFN-$\lambda$ and harvested ileum tissue. Stimulation with small amounts of IFN-$\lambda$ rescued ISG expression in whole tissue (*Figure 2—figure supplement 2*), indicating that that reduction of homeostatic ISG expression upon treatment with ABX is not due to an inability of the intestine to respond to IFN-$\lambda$.

Enteric colonization by bacteria was shown to stimulate systemic type I IFN responses (*Abt et al., 2012*; *Bradley et al., 2019*; *Ganal et al., 2012*; *Steed et al., 2017*; *Stefan et al., 2020*; *Winkler et al., 2020*), so we assessed whether the decreases in ISGs upon ABX treatment or loss of *Ifnlr1* in the ileum were recapitulated in systemic immune tissues. We quantified *Ifit1*, *Stat1*, and *Oas1a* expression in the MLN and the spleen and found that ABX treatment and *Ifnlr1* deletion did not reduce ISGs in these tissues (*Figure 2C*). Although we observe increases in *Ifit1* expression in MLN upon treatment with ABX, these results are not recapitulated by *Stat1* and *Oas1a* expression, and no significant changes in ISG expression were detected in the spleen. Cumulatively, these data indicate that homeostatic ISGs include genes beyond the core signature identified in *Figure 1* (e.g. *Oas1a*), that homeostatic ISGs are present in colonic tissue, and that homeostatic IFN-$\lambda$-stimulated genes are most prominent in enteric tissues.

## Homeostatic ISG expression in the intestine is independent of type I IFN

To determine whether detection of enteric bacteria by TLRs stimulates homeostatic ISGs, we measured tissue ISG expression in mice that were deficient in TRIF or MYD88. Signaling through TRIF results in activation of interferon regulatory factors (IRFs), such as IRF3 and IRF7, that commonly contribute to IFN induction (*Honda et al., 2005a*; *Osterlund et al., 2007*; *Schmid et al., 2010*). Additionally, signaling through MYD88 can aid initiation of IFN expression, through nuclear factor kappa-light-chain-enhancer of activated B cells transcription factor family members (*Osterlund et al., 2007*) and IRF7 (*Honda et al., 2005b*; *Tomasello et al., 2018*). *Ifit1* expression in the ileum and colon of *Trif*[-/-] mice was not significantly different than in WT mice, and *Ifit1* expression was reduced with ABX in *Trif*[-/-] mice (*Figure 3A*). However, we found that mice lacking *Myd88* exhibited tissue-specific decreases in *Ifit1* expression, with significant decreases in the ileum, but not in the colon, relative to WT mice (*Figure 3A*). These data are consistent with a previous report (*Stockinger et al., 2014*) and suggest that signaling through MYD88, but not TRIF, is necessary for homeostatic ISG expression in the ileum, whereas other factors may dominate in the colon.

To expand these findings, we assessed the role of IRF3 and IRF7 transcription factors that are commonly activated downstream of MYD88 and TRIF. We did not observe significant decreases in *Ifit1* expression in the ileum or the colon of *Irf3*[-/-] mice as compared to WT (*Figure 3B*), indicating that IRF3 is not required for homeostatic IFN-$\lambda$ induction. However, we observed a modest (twofold) decrease in *Ifit1* expression in both the ileum and colon of *Irf7*[-/-] mice when compared to WT mice (*Figure 3B*). Although IRF7 is implicated by these data, it does not appear to be strictly required for homeostatic

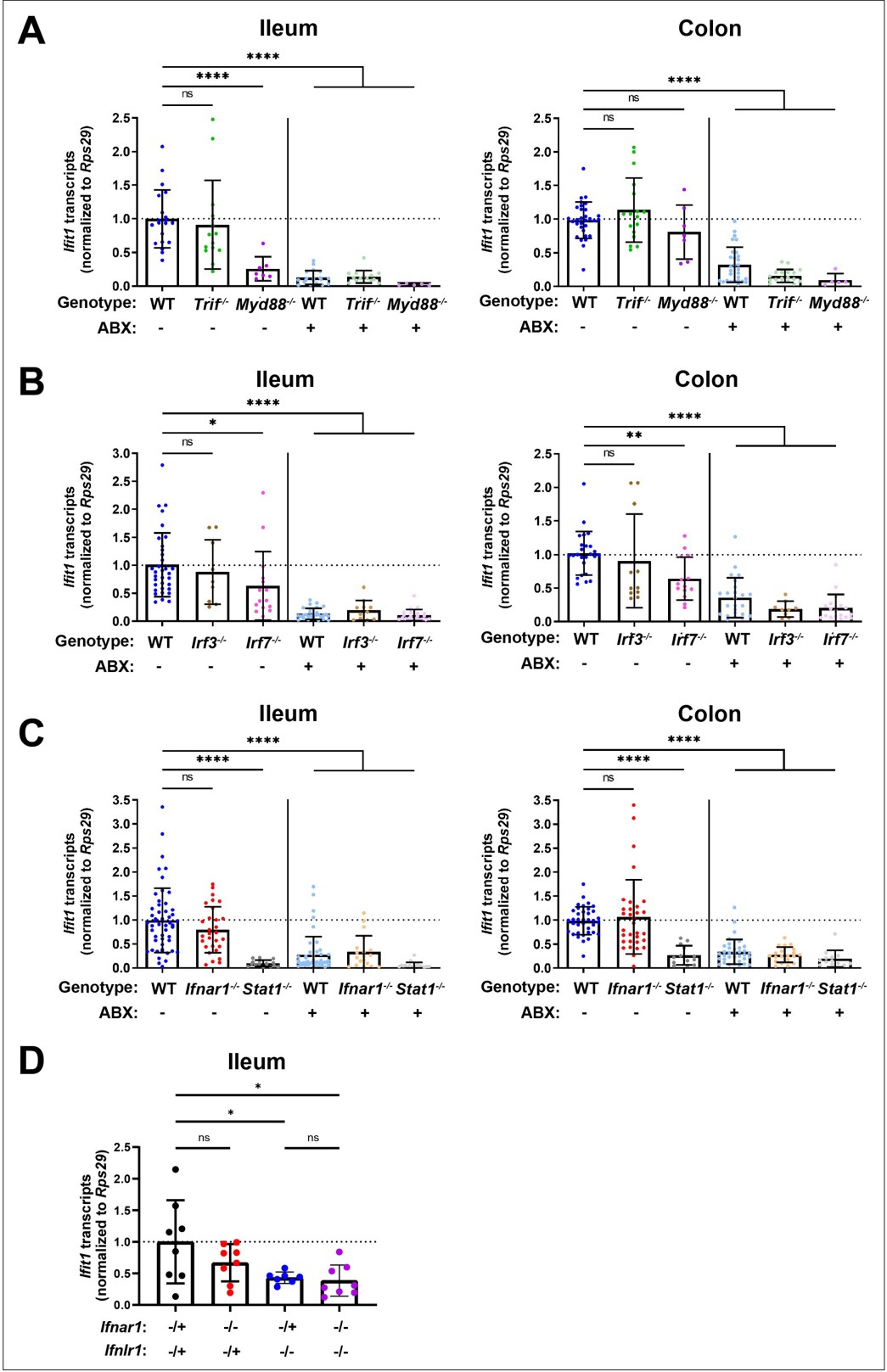

**Figure 3.** Homeostatic interferon-stimulated gene (ISG) expression is independent of type I interferon (IFN). *Ifit1* expression levels were assessed by quantitative PCR (qPCR) from the ileum or colon of (**A**) *Trif⁻/⁻* and *Myd88⁻/⁻*, (**B**) *Irf3⁻/⁻* and *Irf7⁻/⁻*, or (**C**) *Ifnar1⁻/⁻* and *Stat1⁻/⁻* mice treated with or without antibiotics (ABX) normalized to untreated, wild-type (WT) mice. Some WT controls are shared across experiments in (**A–C**). (D) *Ifit1* expression was measured

*Figure 3 continued on next page*

*Figure 3 continued*

by qPCR in ileum tissue of indicated genotypes. Data points represent individual mice and are pooled from at least six independent experiments in (**A–C**) and two independent experiments in (**D**). Statistical significance was determined by one-way ANOVA with Dunnett's multiple comparisons in (**A–C**) and one-way ANOVA with Tukey's multiple comparisons in D, where * = p < 0.05, ** = p < 0.01, *** = p < 0.001, **** = p < 0.0001.

The online version of this article includes the following source data for figure 3:

**Source data 1.** Values and statistical tests of interferon-stimulated gene (ISG) expression in the ileum and colon of genetic knockout mice.

expression of *Ifit1* because expression is further reduced by ABX treatment (***Figure 3B***). These data suggest that IRF7 is not necessary for the homeostatic ISG response to the bacterial microbiota.

Type I and III IFNs stimulate overlapping ISG responses that are both dependent on STAT1. Therefore, to further investigate the contribution of type I IFN signaling to homeostatic ISG expression, we quantified *Ifit1* expression in ileum and colon tissue from *Ifnar1$^{-/-}$* or *Stat1$^{-/-}$* mice. *Ifit1* expression was not significantly different in either ileum or colon tissue of *Ifnar1$^{-/-}$* mice compared to WT (***Figure 3C***). However, *Ifit1* expression was significantly lower in ileum and colon tissue of *Stat1$^{-/-}$* mice compared to *Ifnar1$^{-/-}$* and WT mice. Importantly, treatment of *Stat1$^{-/-}$* mice with ABX did not further reduce *Ifit1* expression, emphasizing the necessity of STAT1 for this homeostatic ISG response.

Lastly, to determine whether type I IFN signaling plays a compensatory role in homeostatic ISG expression in the absence of *Ifnlr1*, we bred mice with heterozygous expression of both *Ifnlr1* and *Ifnar1* (*Ifnlr1$^{-/+}$/Ifnar1$^{-/+}$*) with mice that lack *Ifnlr1* and *Ifnar1* (*Ifnlr1$^{-/-}$/Ifnar1$^{-/-}$*). This breeding scheme produced littermate-matched mice that were *Ifnlr1$^{-/+}$/Ifnar1$^{-/+}$*, *Ifnlr1$^{-/-}$/Ifnar1$^{-/-}$*, and mice singly deficient in *Ifnlr1* (*Ifnlr1$^{-/-}$/Ifnar1$^{-/+}$*) or *Ifnar1* (*Ifnlr1$^{-/+}$/Ifnar1$^{-/-}$*). We found that *Ifit1* expression was not significantly different in *Ifnlr1$^{-/+}$/Ifnar1$^{-/-}$* relative to *Ifnlr1$^{-/+}$/Ifnar1$^{-/+}$* controls in ileal tissue (***Figure 3D***). *Ifit1* expression in *Ifnlr1$^{-/-}$/Ifnar1$^{-/+}$* was significantly lower compared to *Ifnlr1$^{-/+}$/Ifnar1$^{-/+}$* controls, but was not significantly different from *Ifnlr1$^{-/-}$/Ifnar1$^{-/-}$* mice (***Figure 3D***). Cumulatively, these data indicate that homeostatic ISG expression in the intestine is partly dependent on MYD88, and is independent of type I IFN signaling.

## Homeostatic ISG expression in the intestine is restricted to epithelial cells

Given the primarily epithelial expression of *Ifnlr1*, we assessed which compartment of the ileum expresses homeostatic ISGs by isolating a stripped intestinal epithelial fraction and digesting the underlying lamina propria. We assessed ISG expression in these two fractions from WT and *Ifnlr1$^{-/-}$* mice treated with or without ABX. Treatment with ABX or loss of *Ifnlr1* reduced homeostatic ISGs in the IEC fraction, but the lamina propria had low expression of these ISGs relative to the epithelium of untreated WT mice, regardless of treatment or genotype (***Figure 4A***).

IECs express abundant *Ifnlr1*, but other intraepithelial cell types do not (***Hernández et al., 2015***; ***Sommereyns et al., 2008***). To determine whether *Ifnlr1* expression by IECs was required for homeostatic ISG expression, we used mice with IECs that are conditionally deficient in *Ifnlr1* (*Ifnlr1* IEC-cKO) and littermates that retain normal *Ifnlr1* expression (*Ifnlr1$^{flox/flox}$*) (***Baldridge et al., 2017***). *Ifit1* and *Oas1a* expression in the ileum (***Figure 4B***) and colon (***Figure 4C***) of *Ifnlr1$^{flox/flox}$* mice was decreased upon treatment with ABX, consistent with the phenotype observed in WT mice. Conditional deletion of *Ifnlr1* in IECs reduced *Ifit1* and *Oas1a* to a similar extent as the reduction observed in *Ifnlr1$^{-/-}$* animals, above (***Figure 2A–B***). Together, these findings indicate that homeostatic ISGs are dependent on *Ifnlr1* expression by IECs.

## Bacterial microbiota stimulate expression of *Ifnl2/3* by CD45+ cells

A previous study (***Hernández et al., 2015***) noted the presence of IFN-$\lambda$ transcripts (*Ifnl2/3*) at homeostasis in CD45+ cells within the stripped intestinal epithelium, but not in the lamina propria. To extend these findings and to determine whether CD45+ cells in the intestinal epithelium produce IFN-$\lambda$ in response to bacterial microbiota at homeostasis, we enriched cell subsets from epithelial or lamina propria fractions of the ileum for quantification of *Ifnl2/3* by qPCR. We treated WT mice with control, ABX, or stimulation with a synthetic dsRNA analogue (poly I:C). We then magnet-enriched

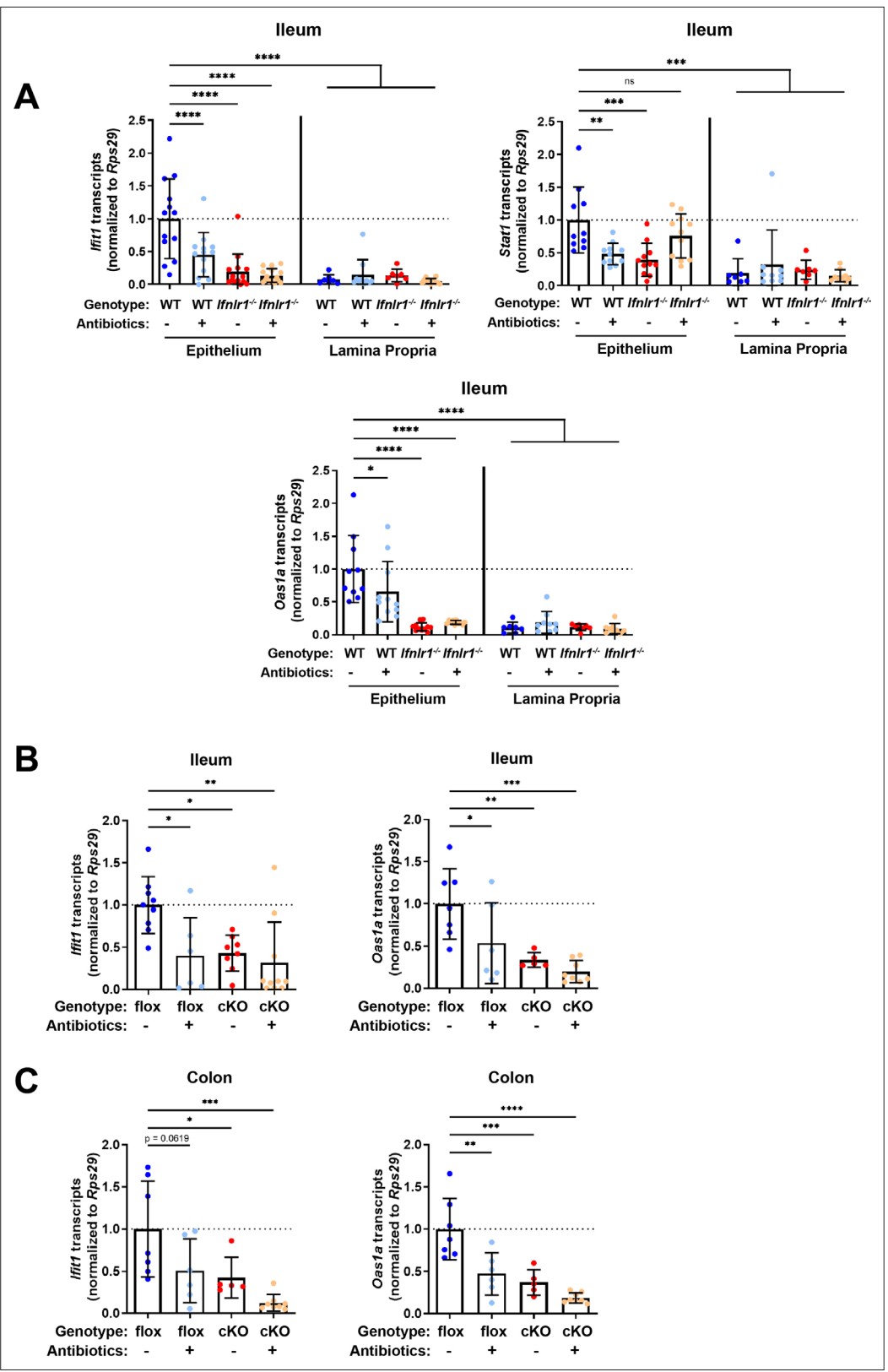

**Figure 4.** Homeostatic interferon-stimulated gene (ISG) expression in the intestine is restricted to epithelial cells. (**A**) *Ifit1*, *Stat1*, and *Oas1a* expression was quantified in stripped epithelial cells or in the lamina propria cells of the ileum. Comparisons were performed between wild-type (WT) and *Ifnlr1⁻/⁻* mice with or without antibiotics (ABX) treatment, and ISG expression was normalized to WT values. (**B–C**) *Ifit1* and *Oas1a* expression from the (**B**) ileum

*Figure 4 continued on next page*

*Figure 4 continued*

or (**C**) colon of mice with conditional presence (flox) or absence (cKO) of *Ifnlr1* in intestinal epithelial cells. Data points represent individual mice and are pooled from three independent experiments in (**A**) and two independent experiments in (**B–C**). Statistical significance was determined by one-way ANOVA with Dunnett's multiple comparisons. * = p < 0.05, ** = p < 0.01, *** = p < 0.001.

The online version of this article includes the following source data for figure 4:

**Source data 1.** Values and statistical tests of interferon-stimulated gene (ISG) expression in the ileum and colon of genetic knockout mice.

EpCAM+ or CD45+ cells from stripped intestinal epithelium or digested lamina propria ileal tissue (*Figure 5—figure supplement 1*). We found that CD45+ cells from the intestinal epithelial and lamina propria fraction expressed detectable *Ifnl2/3* at homeostasis, but EpCAM+ cells did not, which is consistent with Mahlakõiv et al. Furthermore, we found that expression of *Ifnl2/3* in CD45+ cells from the epithelial fraction was significantly reduced with ABX treatment, and CD45+ cells from lamina propria had a non-statistically significant (p = 0.0523) reduction in *Ifnl2/3* expression with ABX treatment (*Figure 5A*). From these data, we conclude that epithelium-associated CD45+ leukocytes are the likely source of homeostatic IFN-$\lambda$ in response to bacterial microbiota, but we do not rule out additional involvement of CD45+ cells in the lamina propria.

Similar to *Ifnl2/3*, we found that CD45+ cells of the epithelial fraction modestly expressed IFN-β transcript (*Ifnb1*) at homeostasis. Additionally, CD45+ cells of the lamina propria robustly expressed *Ifnb1* at homeostasis. However, unlike *Ifnl2/3*, *Ifnb1* was not decreased in mice treated with ABX (*Figure 5B*). These data indicate that *Ifnb1* expression by CD45+ cells in the intestine is less dependent on stimulation by bacterial microbiota relative to *Ifnl2/3*, consistent with the dominant role of IFN-$\lambda$ responses in driving homeostatic ISG expression in the epithelium.

## Homeostatic ISG expression in the small intestine is highly localized

Homeostatic ISG expression in the ileum was of relatively low magnitude when compared to IFN-$\lambda$ treatment (*Figures 1–2*). Therefore, we hypothesized that a low abundance of homeostatic ISG expression would be uniformly distributed between IECs of the intestinal epithelium, and we sought to assess the distribution of the homeostatic ISG, *Ifit1*, using in situ hybridization (RNAscope). Contrary to our hypothesis, RNAscope staining of the ileum from untreated WT mice revealed localized pockets of robust *Ifit1* expression in individual villi rather than ubiquitously low expression throughout the intestinal epithelium (*Figure 6A*). Additionally, *Ifit1* localization was skewed away from the crypt and toward the tips of individual villi within the ileum, and this localization was not specific to *Ifit1* because the distinct ISG *Usp18* co-localized with *Ifit1* (*Figure 6A*). These data indicate that homeostatic ISGs are sporadically expressed in individual villi and are primarily localized to mature enterocytes that are most distally located in villi. We determined that localized ISG expression within individual villi was not due to a localized ability to respond to IFN-$\lambda$ because stimulation with exogenous IFN-$\lambda$ resulted in *Ifit1* expression within all intestinal villi, but not intestinal crypts (*Figure 6—figure supplement 1*). The minimal expression of *Ifit1* within intestinal crypts following exogenous IFN-$\lambda$ treatment suggests that homeostatic ISGs are localized to mature enterocytes because intestinal crypts do not exhibit robust responses to IFN-$\lambda$. Additionally, the non-uniform distribution of homeostatic *Ifit1* expression was ablated in the ileum of mice treated with ABX (*Figure 6B–D*), consistent with a dependency on bacterial microbiota. To determine whether localized *Ifit1* expression was dependent on IEC expression of *Ifnlr1*, we assessed the distribution of *Ifit1* expression in the ilea of littermate Ifnlr1^flox/flox and *Ifnlr1* IEC-cKO mice. We found that the localized *Ifit1* expression observed in untreated WT mice was recapitulated in *Ifnlr1*^flox/flox mice (*Figure 6E*), but these areas of ISG expression were ablated in *Ifnlr1* IEC-cKO mice (*Figure 6F–G*). Lastly, we found that this discrete localization of the homeostatic ISG response is not limited to the ileum, as localized *Ifit1* staining is also observed in the colonic epithelium (*Figure 6—figure supplement 2*). Together, our analyses indicated that homeostatic ISGs are expressed in a highly localized manner within the intestinal epithelium.

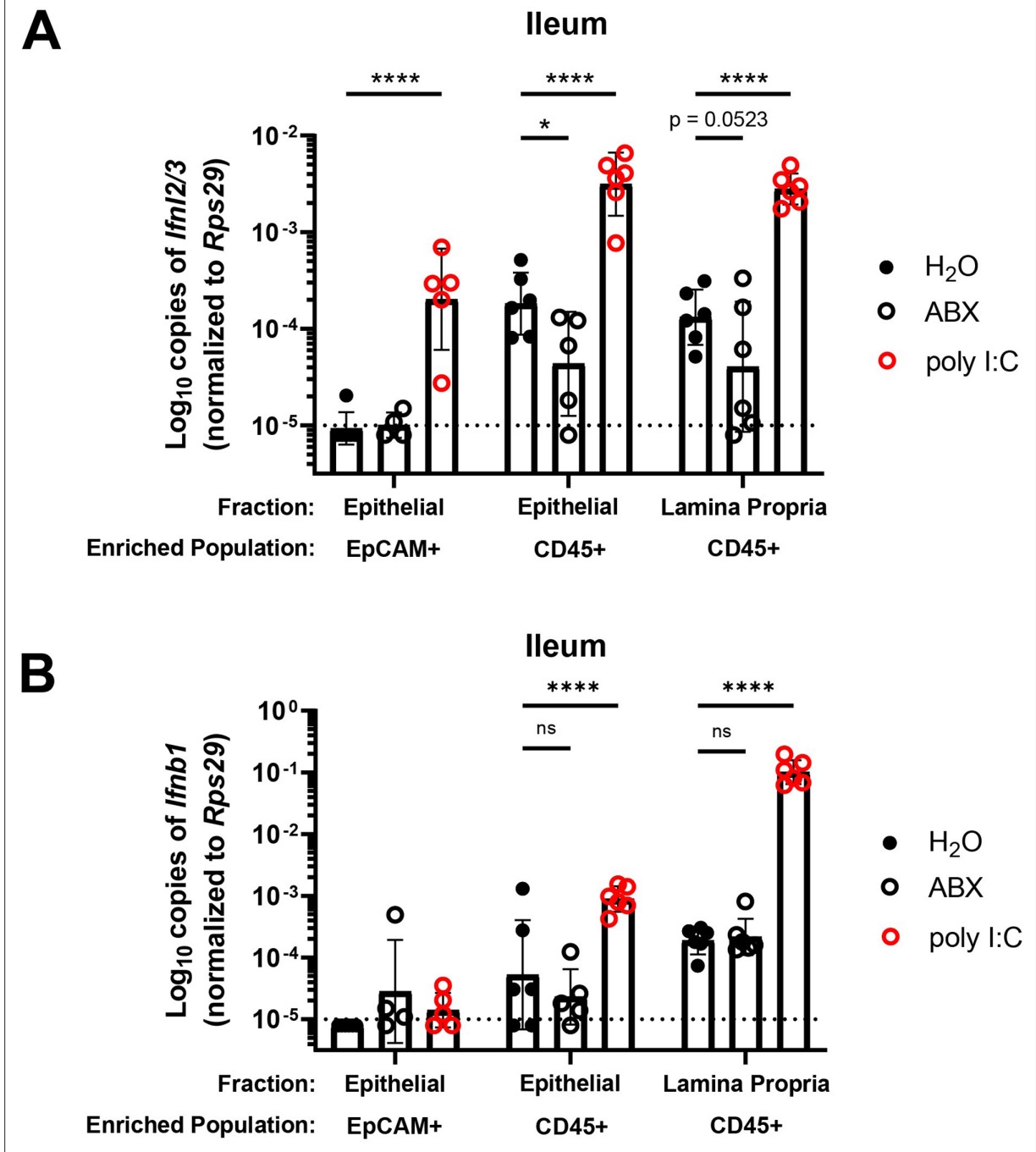

**Figure 5.** Bacterial microbiota stimulate expression of *Ifnl2/3* by CD45+ cells. (A–B) Cellular suspensions from the ileal epithelium and lamina propria were harvested from wild-type (WT) mice treated with H$_2$O, ABX, or stimulated with poly I:C. Resulting cells were enriched for EpCAM-positive and CD45+ cells by magnetic separation. *Ifnl2/3* expression (**A**) and *Ifnb1* expression (**B**) were quantified from each enriched cellular fraction by quantitative PCR (qPCR). Data points represent individual mice and are pooled from two independent experiments. Statistical significance was determined by two-way ANOVA with Dunnett's multiple comparisons, where * = p < 0.05 and **** = p < 0.0001.

The online version of this article includes the following source data and figure supplement(s) for figure 5:

**Source data 1.** Values and statistical tests of interferon expression by CD45+ cells in the ileum.

**Figure supplement 1.** Enrichment of EpCAM+ and CD45+ cells from the intestinal epithelium and lamina propria.

**Figure supplement 1—source data 1.** Values and statistical tests for the enrichment of cells isolated from the ileum.

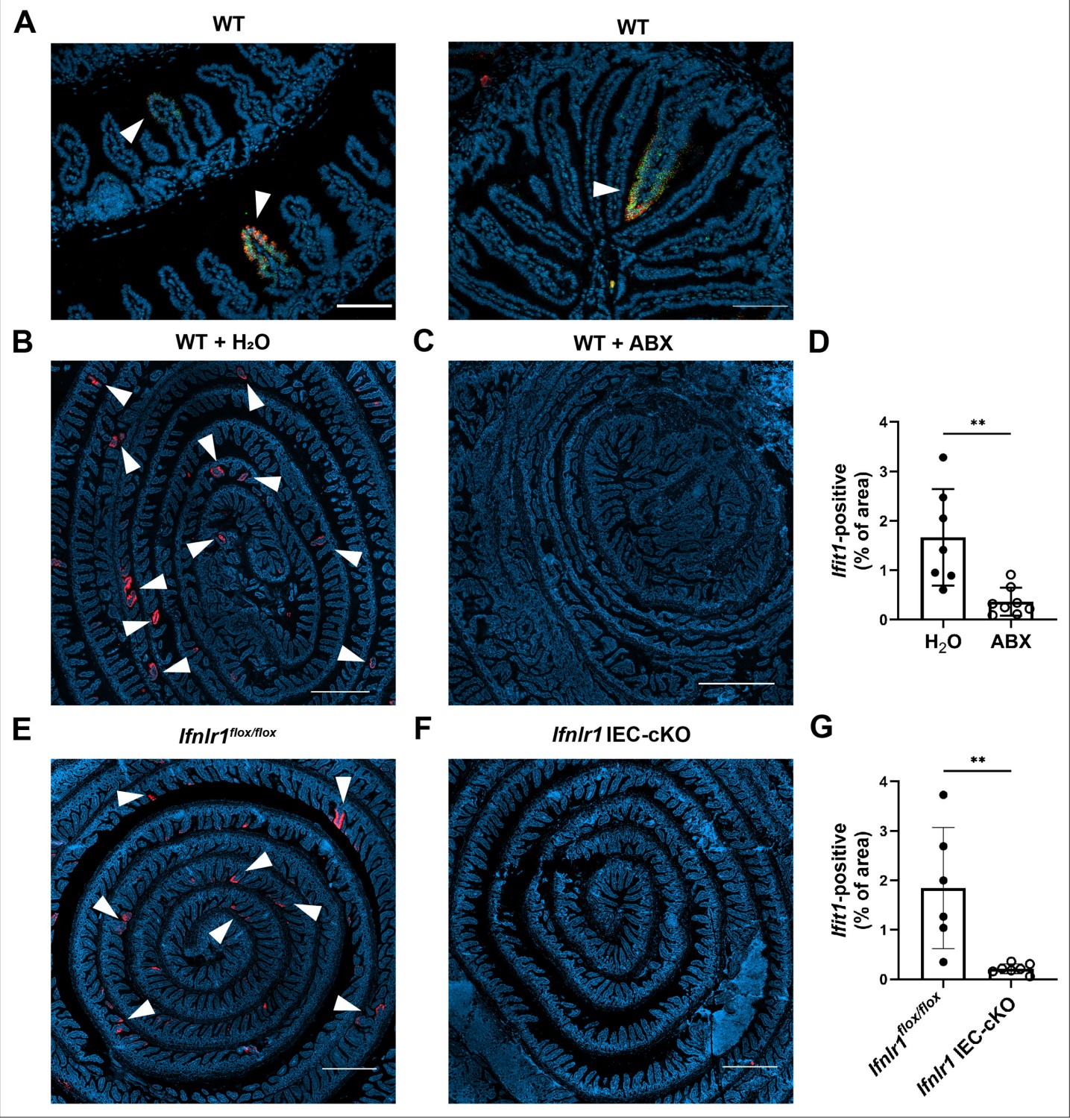

**Figure 6.** Homeostatic interferon-stimulated gene (ISG) expression in the small intestine is highly localized. The ilea of wild-type (WT), *Ifnlr1^flox/flox^*, or *Ifnlr1* intestinal epithelial cell (IEC)-cKO mice were harvested, processed into a Swiss rolls, and stained by in situ hybridization for the ISGs, *Ifit1* (red), and *Usp18* (green), with a DAPI (blue) counterstain.(**A**) Representative high-magnification images of co-localized *Ifit1* (red) and *Usp18* (green) expression in the ileum of WT mice, see arrows. (**B–D**) WT mice treated with $H_2O$ control or antibiotics (ABX) with quantification of *Ifit1* area relative to the total area of the section across replicate mice. (**E–G**) *Ifnlr1^flox/flox^* or *Ifnlr1* IEC-cKO mice at homeostasis with quantification of *Ifit1* area relative to the total area of the section across replicate mice. Scale bar = 100 µm in (**A**) and 500 µm in (**B–G**). Statistical significance was calculated by unpaired t-test where ** = $p < 0.01$. Each data point in (**D**) and (**G**) represents an individual mouse from two independent experiments.

*Figure 6 continued on next page*

*Figure 6 continued*

The online version of this article includes the following source data and figure supplement(s) for figure 6:

**Source data 1.** Values and statistical tests of *Ifit1* area in the ileum of treated and genetic knockout mice.

**Figure supplement 1.** The entire intestinal epithelium is responsive to interferon-lambda (IFN-λ).

**Figure supplement 1—source data 1.** Values and statistical tests of interferon-stimulated gene (ISG) expression in the ileum of interferon-lambda (IFN-λ)-treated mice.

**Figure supplement 2.** Homeostatic interferon-stimulated gene (ISG) expression is also highly localized in the colon.

**Figure supplement 2—source data 1.** Values and statistical tests of *Ifit1* area in the ileum of genetic knockout mice.

## Mature enterocytes express homeostatic ISGs in public single-cell datasets from mouse and human

To determine the extent of conservation of homeostatic ISG expression by IECs, we performed orthogonal analyses of publicly available scRNA-seq datasets from mouse (*Haber et al., 2017*) and human (*Elmentaite et al., 2020*) IECs. Recently, Haber et al. published a large scRNA-seq dataset that profiled sorted IECs from the small intestine of specific pathogen-free mice and defined 15 distinct IEC subtypes (*Haber et al., 2017*). We analyzed IECs from this dataset, and found that of the 21 homeostatic ISGs identified in *Figure 1*, 19 were present in the Haber et al. single-cell dataset. We determined the percentage of each epithelial cell subtype that expresses each individual homeostatic ISG, and generated a heatmap with hierarchical clustering to group IEC subtypes that have similar ISG expression patterns (*Figure 7A*). Homeostatic ISGs were predominantly expressed in mature enterocyte subtypes, which clustered separately from crypt-resident progenitor IECs such as transit amplifying (TA) cells and stem cells (*Figure 7A*). To compare homeostatic ISG expression between polar extremes of the crypt-villus axis, we grouped IEC subtypes that represented enterocytes (mature enterocyte cells) and crypt-associated cells (TA cells and stem cells) to compare the overall proportions of these cells with homeostatic ISG expression. We found that a significantly higher percentage of enterocytes express homeostatic ISGs than crypt-associated cells, but that these homeostatic ISGs were expressed in a relatively small proportion of enterocytes (<20%) (*Figure 7B*). Notably, *Ifit1* (highlighted in red) was present in ~5% of enterocytes by scRNA-seq, which is consistent with our observation of 1–4% *Ifit1*-positive area by imaging the mouse small intestine (*Figure 6B–G*). Furthermore, the relative absence of ISG-positive crypt-associated cells in this scRNA-seq data is consistent with our observation that intestinal crypts lacked *Ifit1* and *Usp18* expression by imaging.

We expanded our investigation to an scRNA-seq dataset from the ileum of healthy, human, pediatric patients that was previously described (*Elmentaite et al., 2020*). IECs from this dataset were previously clustered by Elmentaite et al. and annotated as: enterocytes, early enterocytes, tuft cells, enteroendrocrine cells, BEST4 enterocytes, goblet cells, TA cells, and crypt. For our analysis of homeostatic ISG expression, we excluded IEC subtypes with fewer than 20 constituent cells, which retained five annotated groups: enterocytes, early enterocytes, goblet cells, TA cells, and crypt. Of the 21 homeostatic ISGs identified in *Figure 1*, 14 orthologous human genes were present in these data. Similar to analysis in *Figure 7A*, we determined the percentage of each group that expressed each individual homeostatic ISG and generated a heatmap of these data with hierarchical clustering to group IEC subtypes that have similar ISG expression patterns (*Figure 7C*). Enterocyte, early enterocyte, and goblet subtypes clustered separately from TA cells and crypt, with the highest proportion of homeostatic ISGs being present in the enterocyte subtype (*Figure 7C*). As with mouse IEC data, above, we grouped annotated cells that localize in the crypt (TA cells and crypt) and compared overall proportions of homeostatic ISG expression with the mature enterocyte subtype (*Figure 7D*). Similar to mice, homeostatic ISGs were present in significantly more enterocytes than crypt-associated cells, and most homeostatic ISGs in this human dataset were present in a relatively small proportion of cells (<20%). These human data suggest that ISGs may be present in a small proportion of IECs from healthy, human ileal tissue at homeostasis and may share the localization observed in our murine analyses. Our analysis of a murine scRNA-seq dataset support our previous observations that homeostatic ISGs are not ubiquitously expressed throughout the intestinal epithelium; rather, they are expressed in a minority of IECs and skewed toward mature enterocytes along the crypt-villus axis. Our analysis of a human scRNA-seq dataset are consistent with observations in the murine model, though future studies will be required to definitively address the existence of homeostatic ISGs in human tissue.

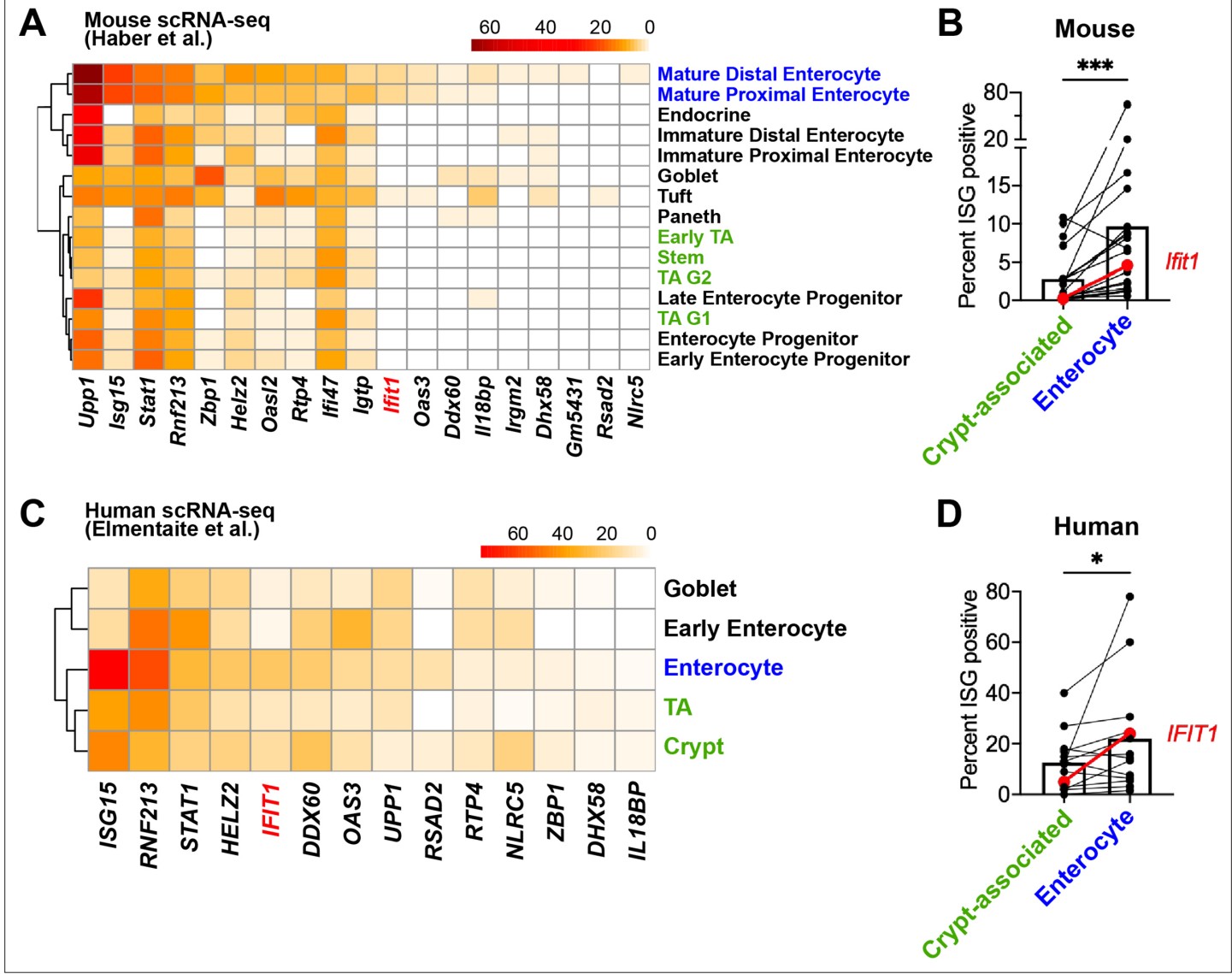

**Figure 7.** Mature enterocytes express homeostatic interferon-stimulated genes (ISGs) in public single-cell datasets from mouse and human. (**A–B**) A mouse intestinal epithelial cell (IEC) single-cell transcriptional dataset (***Haber et al., 2017***) (GSE92332) was analyzed to determine the percentage of each epithelial cell subtypes that express homeostatic ISGs. (**A**) Heatmap depicting the proportion of each epithelial cell type expressing 19 of the 21 homeostatic ISGs identified in ***Figure 1***. (**B**) Enterocyte subtypes (blue text) and crypt-resident progenitor subtypes (green text) cells were grouped and the percentage of cells that express each homeostatic ISG was compared. (**C–D**) A human IEC single-cell transcriptional dataset (***Elmentaite et al., 2020***) (E-MTAB-8901) was analyzed for the percentage of epithelial cell subtypes that express homeostatic ISGs. (**C**) A heatmap depicting the percentage of IEC subtypes that express human orthologs of murine homeostatic ISGs identified in ***Figure 1***. (**D**) The mature enterocyte subtype (blue text) and crypt-resident progenitor subtypes (green text) were grouped and the percentage of cells that express each homeostatic ISG was compared. Lines in (**B**) and (**D**) link paired ISGs in each IEC subset. Statistical significance in (**B**) and (**D**) was calculated by Wilcoxon test where * = p < 0.05, and *** = p < 0.001.

The online version of this article includes the following source data for figure 7:

**Source data 1.** Values and statistical tests of the percentage of cells expressing individual interferon-stimulated genes (ISGs) in single-cell RNA sequencing datasets.

## Assessing the effect of peroral bacterial products on homeostatic ISGs in ABX-treated mice

To further define the relationship between bacteria and homeostatic ISGs, we assessed whether oral administration of fecal contents or purified LPS (a bacterial MAMP and TLR4 agonist) could restore localized ISG expression in mice with depleted bacterial microbiota. Control groups of *Ifnlr1*^*flox/flox*^

mice with conventional microbiota (*Figure 8A*) retained a localized *Ifit1* expression pattern, whereas ABX-treated *Ifnlr1*^flox/flox^ mice (*Figure 8B*) and *Ifnlr1* IEC-cKO mice (*Figure 8C*) lacked *Ifit1* expression. Oral LPS administered to conventional *Ifnlr1*^flox/flox^ mice did not significantly alter the distribution or frequency of localized *Ifit1* expression (*Figure 8D*). However, localized *Ifit1* expression was visible in 4/12 ABX-treated *Ifnlr1*^flox/flox^ mice administered LPS (*Figure 8E and H–I*). In ABX-treated *Ifnlr1*^flox/flox^ mice treated with fecal transplant of conventional microbes, localized expression of *Ifit1* was visible in 4/8 mice (*Figure 8F and H–I*). Importantly, we did not observe localized *Ifit1* expression in *Ifnlr1* IEC-cKO mice following LPS administration (*Figure 8G*), indicating that LPS-stimulated *Ifit1* depends on IEC expression of *Ifnlr1*.

We noted that the visibility of localized *Ifit1* signal following LPS administration or fecal transplant appeared largely binary (i.e. present or absent) in our imaging data (*Figure 8E and F*, representative *Ifit1*-positive and *Ifit1*-negative images). Using quantification of *Ifit1* area (*Figure 8H*), we stratified mice into *Ifit1*-postive and *Ifit1*-negative groups (*Figure 8I–J*) based on a cutoff set at the maximal *Ifit1* area value of *Ifnlr1* IEC-cKO mice (dashed line in *Figure 8H*). Results of this unbiased stratification were consistent with visible *Ifit1* staining and indicated that 8/8 conventional *Ifnlr1*^flox/flox^ mice, 0/8 ABX-treated control mice, 4/12 LPS-treated mice, and 4/8 fecal transplant mice were *Ifit1*-positive (*Figure 8I–J*). Statistical analysis by Fisher's exact test indicated that LPS administration non-significantly increased (p = 0.1022) the proportion of mice that were *Ifit1*-positive, whereas fecal reconstitution of ABX-treated mice significantly increased the likelihood of these mice being *Ifit1*-positive (*Figure 8J*). Importantly, mice that received fecal transplant had full restoration of 16S gene copies (*Figure 8K*) despite only 4/8 having homeostatic *Ifit1* expression. These findings suggest that reconstitution of the homeostatic ISG signal by fecal transfer has incomplete penetrance at this timepoint, underscoring the incomplete presence of localized *Ifit1* expression (4/12) in ABX-treated mice administered peroral LPS.

To corroborate and extend these findings, we performed orthogonal analyses of *Ifit1*, *Stat1*, and *Oas1a* expression in ileum tissue of WT mice treated with ABX followed by fecal transplant, LPS administration, administration of the TLR5 agonist: flagellin, or administration of TLR9 agonist: CpG DNA (*Figure 8—figure supplement 1*). Similar to imaging data, these qPCR data exhibited high variance. However, stratification of tissues into positive and negative for each ISG indicated that peroral administration of LPS to ABX-treated mice increased the proportion of ISG-positive tissues by 21–36% (*Ifit1*: p = 0.065; *Stat1*, p < 0.05; *Oas1a*, p < 0.05). Additionally, peroral administration of flagellin significantly increased the proportion of ISG-positive mice by 25–44% (*Ifit1*: p = 0.052; *Stat1*, p < 0.01; *Oas1a*, p < 0.05), whereas CpG DNA did not significantly increase the proportion of ISG-positive mice (*Figure 8—figure supplement 1*). Together, these data suggest that LPS and flagellin are sufficient to stimulate homeostatic ISG expression in a significant proportion of ABX-treated mice. The ability of multiple PRR ligands to stimulate homeostatic ISGs suggests that exposure to a variety of bacterial MAMPs is the basis for localized, homeostatic ISG expression.

## The homeostatic IFN-λ response preemptively protects IECs from murine rotavirus infection

To assess the capacity of homeostatic ISGs to protect IECs from viral infection, we utilized infection with murine rotavirus (mRV), an IEC-tropic pathogen. Prior studies of rotaviruses have identified viral immune evasion genes that block IFN induction through multiple mechanisms (*Arnold et al., 2013*). However, we reasoned that preexisting ISG expression stimulated by the bacterial microbiome at homeostasis may preemptively protect IECs from the infection before viral gene expression is initiated. To determine the role of an epithelial IFN-λ response over the course of mRV infection in adult mice, we monitored daily shedding of viral genomes in the stool of *Ifnlr1*^flox/flox^ mice and *Ifnlr1* IEC-cKO littermates. We first detected mRV shedding in the stool on day 2 after inoculation and, at this early timepoint, *Ifnlr1* IEC-cKO mice shed 20-fold more mRV genomes into their stool than *Ifnlr1*^flox/flox^ mice (*Figure 9A*). However, at the peak of viral shedding between days 3 and 5, there were no significant differences between *Ifnlr1*^flox/flox^ and *Ifnlr1* IEC-cKO littermates (*Figure 9A*). This similarity at peak of viral shedding was consistent with an ability of mRV to evade the host IFN response once infection is established. Together, these findings suggest that *Ifnlr1* IEC-cKO mice have defects in protection against initiation of mRV infection and that the protective capacity of endogenous IFN-λ signaling against mRV is primarily prophylactic in nature.

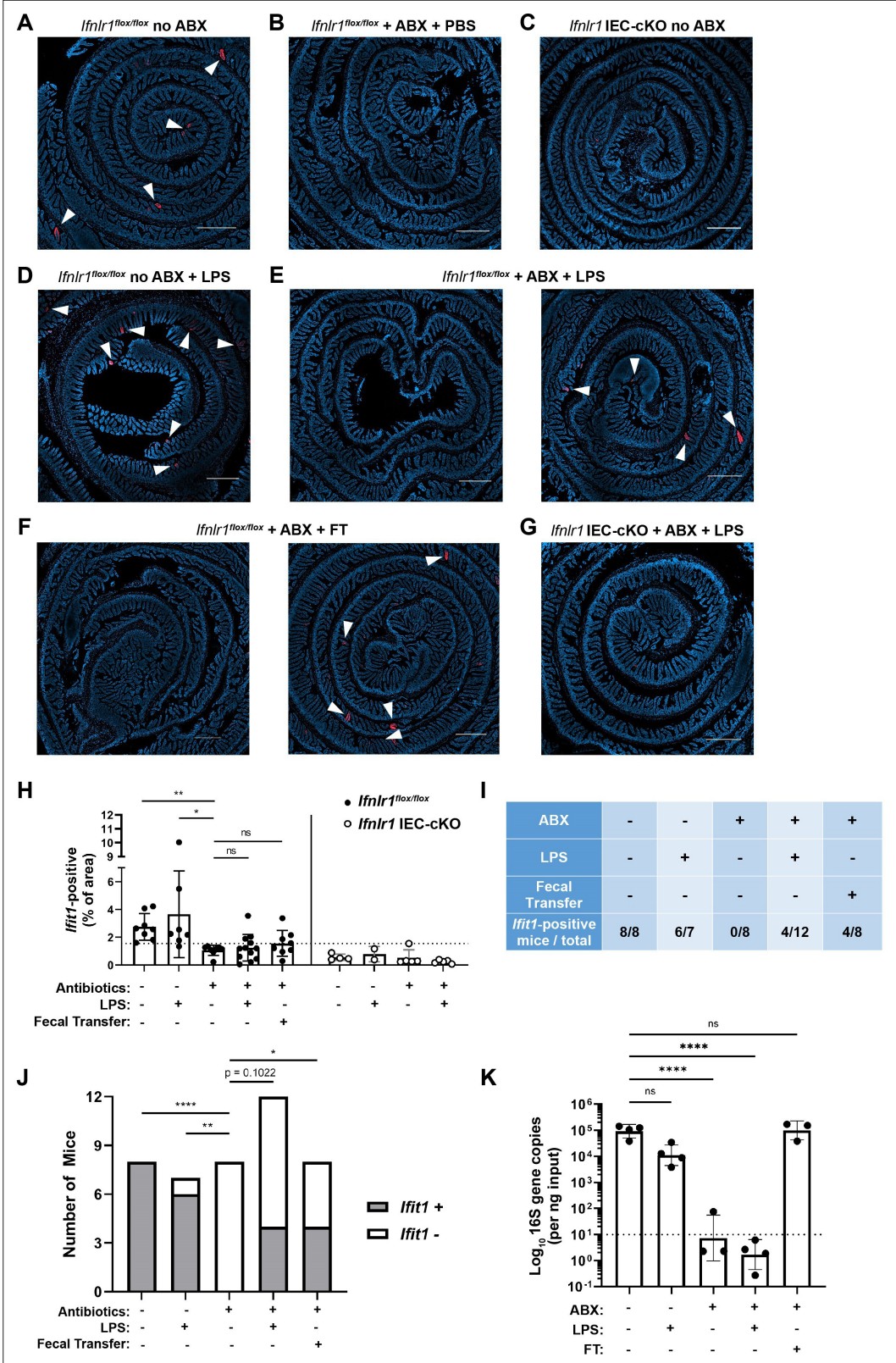

**Figure 8.** Assessing the effect of peroral bacterial products on localized homeostatic interferon-stimulated genes (ISGs) in antibiotics (ABX)-treated mice. The ilea of treated wild-type (WT), *Ifnlr1^flox/flox^*, and *Ifnlr1* intestinal epithelial cell (IEC)-cKO mice were harvested, processed into Swiss rolls, and stained by in situ hybridization for the ISG, *Ifit1* (red), with a DAPI (blue) counterstain. (**A–C**) Representative images from *Ifnlr1^flox/flox^* mice treated with

*Figure 8 continued on next page*

*Figure 8 continued*

H₂O control followed by PBS stimulation (**A**), ABX followed by PBS stimulation (**B**), or from *Ifnlr1* IEC-cKO mice (**B**). (D) Representative images of *Ifnlr1ᶠˡᵒˣ/ᶠˡᵒˣ* mice treated with H₂O control followed by lipopolysaccharide (LPS) stimulation. (**E–F**) Two representative images of *Ifnlr1ᶠˡᵒˣ/ᶠˡᵒˣ* mice treated with ABX followed by LPS stimulation (**E**) or ABX followed by fecal transplantation (**F**). (G) A representative image of *Ifnlr1* IEC-cKO mice treated with ABX followed by LPS stimulation. (**H**) Quantification of *Ifit1* area relative to the total area of each tissue section with a dashed line at the highest *Ifnlr1* IEC-cKO value. The proportion of *Ifit1*-positive mice (above dashed line) and *Ifit1*-negative mice (below dashed line) are tabulated (**I**) and graphed (**J**) for *Ifnlr1ᶠˡᵒˣ/ᶠˡᵒˣ* mice of each condition. (**K**) rDNA was isolated from the luminal contents of mice at endpoint harvest. 16S gene copies were assessed by quantitative PCR (qPCR) and normalized to input. Limit of detection: dashed line. Where depicted, scale bar = 500 μm. Data points represent individual mice and are pooled from four independent experiments in (**A–J**) and from two independent experiments in (**K**). Statistical significance was determined by Kruskal-Wallis with Dunn's multiple comparisons in (**H**), by one-way ANOVA with Dunnett's multiple comparisons in (**K**), and by Fisher's exact tests in (**J**) where * = p < 0.05, ** = p < 0.01, and **** = p < 0.0001.

The online version of this article includes the following source data and figure supplement(s) for figure 8:

**Source data 1.** Values and statistical tests of *Ifit1* area, the proportion of *Ifit1*-positive cells, and 16S gene copies.

**Figure supplement 1.** Assessing the effect of peroral bacterial products on homeostatic interferon-stimulated genes (ISGs) in antibiotics (ABX)-treated mice.

**Figure supplement 1—source data 1.** Values and statistical tests of interferon-stimulated gene (ISG) expression and the proportion of *Ifit1*-positive cells.

To more stringently assess the capacity of *Ifnlr1* to protect IECs against the earliest stages of mRV infection, we inoculated mice with 5000 SD50 (50% shedding dose) of mRV to maximize the likelihood of uniform viral exposure throughout the intestine. At 24 hr post-inoculation, we quantified mRV genomes in the epithelial fraction and the proportion of infected IECs (live, EpCAM-positive, CD45-negative, and mRV-positive) by flow cytometry. At 24 hr post-inoculation, we found that *Ifnlr1* IEC-cKO mice had 20-fold more mRV genomes than *Ifnlr1ᶠˡᵒˣ/ᶠˡᵒˣ* mice (**Figure 9B**). In addition, we found that a threefold greater proportion of IECs were infected with mRV in *Ifnlr1* IEC-cKO mice than *Ifnlr1ᶠˡᵒˣ/ᶠˡᵒˣ* mice at 24 hr post-infection (**Figure 9C–F**). However, the median fluorescence intensity (MFI) of mRV antigen was equivalent in mRV-infected IECs from *Ifnlr1ᶠˡᵒˣ/ᶠˡᵒˣ* and *Ifnlr1* IEC-cKO mice (**Figure 9G**) and the MFI of mRV did not correlate (r² = 0.0003) with the percentage of infected IECs (**Figure 9H**). This equivalent mRV antigen burden in infected cells from *Ifnlr1ᶠˡᵒˣ/ᶠˡᵒˣ* and *Ifnlr1* IEC-cKO mice in combination with the lack of correlation between antigen burden and percentage of infected cells suggests that the protective role of *Ifnlr1* is to prevent infection of IECs rather than to limit replication within IECs after they are infected. We assessed an earlier timepoint at 12 hr post-inoculation and observed similar trends toward an increased proportion of IEC infection in *Ifnlr1* IEC-cKO relative to *Ifnlr1ᶠˡᵒˣ/ᶠˡᵒˣ* littermates, but the extent of infection was 10- to 100-fold lower and near the limit of detection (**Figure 9—figure supplement 1**). Thus, the 12 and 24 hr timepoints capture the earliest detectable infection of IECs by mRV, and this early infection is significantly reduced by IFN-λ signaling.

To further contextualize the localization of ISGs and mRV-infected cells over time, we performed in situ hybridization for mRV genomes and *Ifit1* in the ileum of WT mice at 12, 24, and 96 hr post-inoculation. Quantification of co-staining for *Ifit1* in mRV-infected cells indicated that a minority (~30%) were *Ifit1*+ at 12 or 24 hr post-inoculation, whereas a majority (~63%) of mRV-infected cells were *Ifit1*+ at 96 hr post-inoculation (**Figure 9I–J**, **Figure 9—figure supplement 2**). These co-staining data are consistent with early mRV evasion of IFN responses within infected IECs at the same timepoints that we observed increased infection of *Ifnlr1* IEC-cKO mice (**Figure 9B–F**, **Figure 9—figure supplement 1**). Furthermore, we found mRV inoculation did not increase the area of *Ifit1* expression in the intestine at 12–24 hr post-inoculation. However, at 96 hr post-inoculation, we found a significant increase in epithelial *Ifit1* expression in the ileum that coincided with increased viral genomes and antigen (**Figure 9—figure supplement 3**), suggesting that *Ifit1* expression at 12 and 24 hr post-inoculation is primarily due to the preexisting homeostatic response rather than in response to mRV infection. Therefore, we propose that homeostatic ISGs stimulated by *Ifnlr1* expression on IECs play an early protective role against viral infection that preempts viral IFN evasion mechanisms.

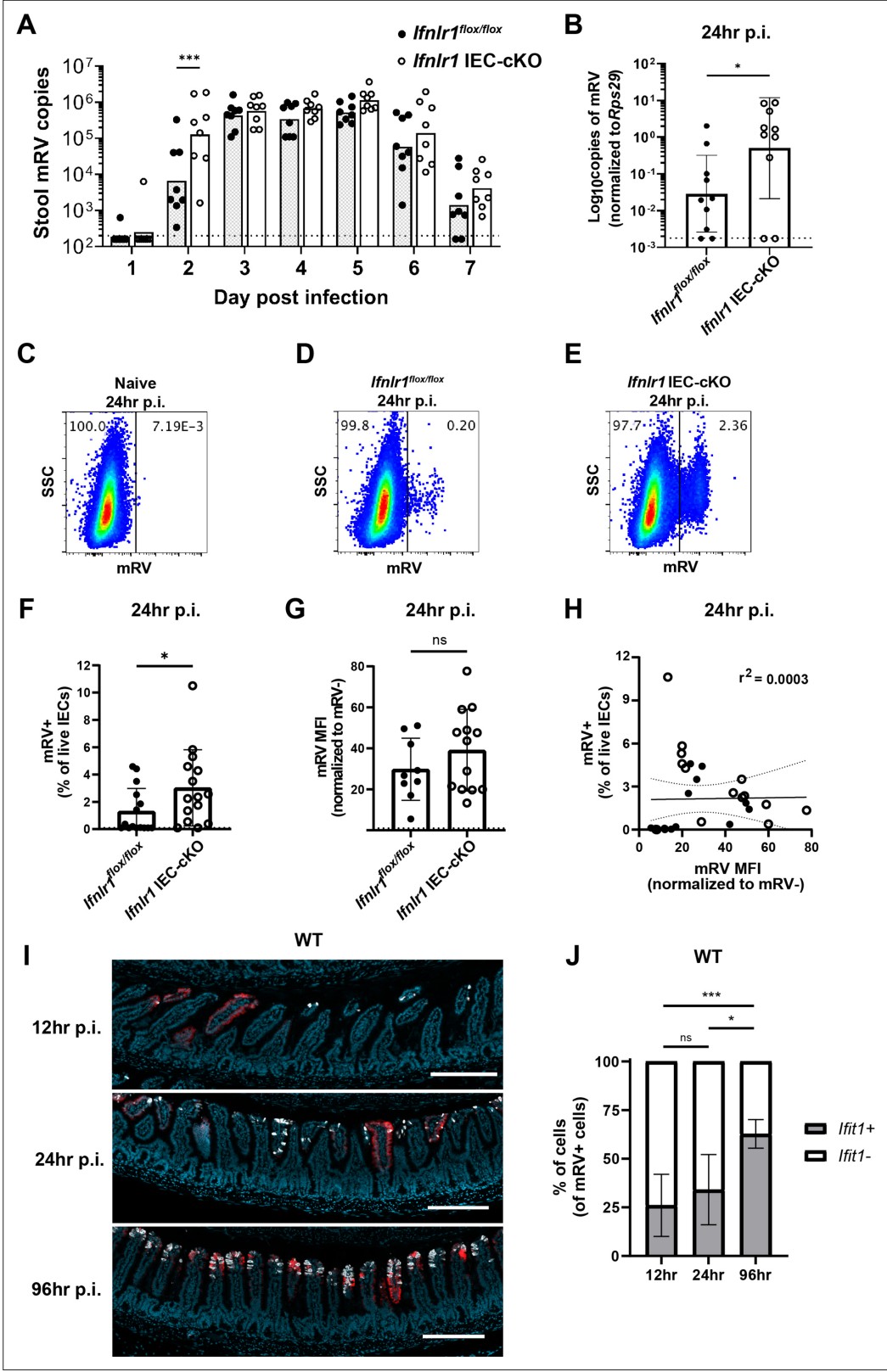

**Figure 9.** The homeostatic interferon-lambda (IFN-$\lambda$) response preemptively protects intestinal epithelial cells (IECs) from murine rotavirus (mRV) infection. *Ifnlr1*<sup>flox/flox</sup> and *Ifnlr1* IEC-cKO mice were infected with 100 SD50 (**A**) or 5000 SD50 (**B–H**) of mRV and stool (**A**) or stripped ileal IECs (**B–G**) were assessed for mRV infection. (**A**) Timecourse of mRV genome copies detected in the stool of *Ifnlr1*<sup>flox/flox</sup> and *Ifnlr1* IEC-cKO mice by quantitative PCR (qPCR).

*Figure 9 continued on next page*

*Figure 9 continued*

(**B–G**) Mechanically stripped IEC fractions were analyzed by qPCR for mRV genome copies (**B**) or flow cytometry for mRV antigen-positive IECs (**C–G**). Representative flow cytometry plots of naïve (**C**), *Ifnlr1*$^{flox/flox}$ (**D**), and *Ifnlr1* IEC-cKO (**E**) mice infected with mRV with quantification in (**F**). (**G**) The median fluorescence intensity (MFI) of mRV antigen in infected cells relative to uninfected cells. (**H**) A linear correlation plot of mRV+ cells and mRV MFI with 95% confidence intervals (dashed lines). (**I–J**) Wild-type (WT) mice were infected with 5000 SD50 of mRV and the ilea were processed into Swiss rolls and stained by in situ hybridization for the interferon-stimulated gene (ISG), *Ifit1*, or mRV. The percentage of mRV-infected cells that were *Ifit1*-positive and *Ifit1*-negative were determined. Additional representative images for (**I–J**) are depicted in ***Figure 9—figure supplement 2***. Where designated in (**A–G**), dashed lines = limit of detection (LOD) as set by naïve mice. Data points represent individual mice and are pooled from two to three independent experiments. Statistical significance was determined by two-way ANOVA with Sidak's multiple comparisons (**A**), by Mann Whitney (**B, F**), unpaired t-test (**G**), or Kruskal-Wallis with Dunn's multiple comparisons (**J**), where * = p < 0.05 and *** = p < 0.001.

The online version of this article includes the following source data and figure supplement(s) for figure 9:

**Source data 1.** Values and statistical tests of murine rotavirus (mRV) genomes, percentages of cells infected, median fluorescence intensity (MFI), and % of cells mRV+ cells co-staining with *Ifit1* expression.

**Figure supplement 1.** The homeostatic interferon-lambda (IFN-$\lambda$) response preemptively protects intestinal epithelial cells (IECs) from murine rotavirus (mRV) infection.

**Figure supplement 1—source data 1.** Values and statistical tests of murine rotavirus (mRV) genomes, percentages of cells infected, and median fluorescence intensity (MFI).

**Figure supplement 2.** Murine rotavirus (mRV) and *Ifit1* co-incidence.

**Figure supplement 2—source data 1.** Percentage values of murine rotavirus (mRV)+ cells co-staining with *Ifit1* expression.

**Figure supplement 3.** Murine rotavirus (mRV) infection increases the distribution of *Ifit1* expression at late times post-inoculation.

**Figure supplement 3—source data 1.** Values and statistical tests of *Ifit1* area, murine rotavirus (mRV) genomes, and percentage of mRV-infected cells.

## Discussion

Here, we report that bacterial microbiota induce an enteric IFN-$\lambda$ response in IECs at homeostasis (***Figures 1–4***). Although there are previous reports of basal, non-receptor-dependent ISG expression in immortalized and primary cell lines (***Hakim et al., 2018***), the homeostatic response that we report here is dependent on the IFN-$\lambda$ receptor. Furthermore, we find this response is independent of type I IFN signaling (***Figure 3***), indicating that IFN-$\lambda$ signaling plays a dominant and active role in the gastrointestinal epithelium. We also find minimal changes in ISG expression within the spleen and MLN after ABX treatment, suggesting that homeostatic ISGs are predominantly expressed in the intestine. These findings differ slightly from prior descriptions of systemic type I IFN responses that are dependent on bacterial microbiota (***Abt et al., 2012***; ***Bradley et al., 2019***; ***Ganal et al., 2012***; ***Steed et al., 2017***; ***Stefan et al., 2020***; ***Winkler et al., 2020***). However, differences in the specific tissues and cell types analyzed make it difficult to draw direct comparisons across these studies. Therefore, we conclude that homeostatic ISGs are substantially present in IECs, but do not dispute the prior findings that relatively low homeostatic ISG expression induced by type I IFN plays an important role in extra-intestinal tissues and non-epithelial cell types.

A prior study by Mahlakõiv et al. noted the presence of *Ifnl2/3* transcripts in CD45+ cells within the stripped intestinal epithelium, but not the lamina propria, at homeostasis. We have confirmed these findings and have extended them to show that this homeostatic expression of *Ifnl2/3*, but not *Ifnb1*, is dependent on bacterial microbiota (***Figure 5***). However, we have been unable to detect IFN-$\lambda$ transcripts by RNA scope in situ hybridization in mice at homeostasis, consistent with recently published data (***Ingle et al., 2021***). This suggests that the production of IFN-$\lambda$ is below the limit of detection by imaging or highly transient in nature. It is also unclear which CD45+ cell type produces homeostatic IFN-$\lambda$. Swamy et al. showed that T cell receptor stimulation led intraepithelial lymphocytes to produce IFN-$\lambda$ (***Swamy et al., 2015***) and Mahlakõiv et al. suggested that the primary producers of IFN-$\lambda$ at steady state are intraepithelial lymphocytes due to the abundance of intraepithelial lymphocytes in the epithelial fraction. However, myeloid cells also reside near the intestinal epithelium and

can sample luminal contents by various mechanisms (*Chieppa et al., 2006*; *McDole et al., 2012*; *Niess et al., 2005*). Although the cell type that produces homeostatic IFN-$\lambda$ is unknown, enrichment for *Ifnl2/3* in CD45+ cells within the intestinal epithelial fraction suggests that proximity to bacterial stimuli may be a primary determinant in this response. We suggest that these cells may be actively surveying the intestinal epithelium to detect bacterial MAMPs. The specific CD45+ cell types responsible for producing homeostatic IFN-$\lambda$ will be a topic of interest for future studies.

We initially anticipated that the distribution of homeostatic ISGs would be uniform among IECs. Instead, we found that this IFN-$\lambda$ response is highly localized within enteric tissues. Homeostatic ISGs are observed in a minority of small intestinal villi and are primarily present in mature epithelium toward the villus tips (*Figure 6*). Likewise, homeostatic ISGs are present within patches of the mature epithelium in the colon (*Figure 6—figure supplement 2*). The surprising finding of localized ISGs in the ileum are supported by analysis of independently generated scRNA-seq datasets from mouse and human small intestinal IECs that depict expression of homeostatic ISGs in a minority of cells with predominant expression in mature enterocytes (*Figure 7*).

The basis for localized ISG expression is unknown; however, it may reflect the distribution of cells capable of sensing bacterial microbiota, distinct microenvironments within the gastrointestinal tract, or qualitative differences in bacterial colonization. Given the results of our data in *Figure 8* and *Figure 8—figure supplement 1*, we suggest that LPS administration, flagellin administration, or fecal transplant can partially restore the expression of homeostatic ISGs in the small intestine. This interpretation supports the concept that localized ISG-positive regions may be uniquely exposed or responsive to a variety of bacterial MAMPs. Indeed, we find that MYD88 is required for WT levels of homeostatic ISG expression in the small intestine (*Figure 3*). However, MYD88 is dispensable for homeostatic ISG expression in the colon, and TRIF is not required in either small intestine or colon (*Figure 3*). These data suggest that the bacterial microbiota broadly stimulates homeostatic IFN-$\lambda$ through multiple, redundant PRRs. Furthermore, the presence of localized *Ifit1* expression in ABX-treated mice following LPS administration (*Figure 8*) suggests that localization is an intrinsic property of homeostatic ISG stimulation and is not likely to be due to qualitative differences in bacterial colonization.

ISG localization may be indicative of regional differences in access of luminal bacterial MAMPs to IFN-$\lambda$-producing cells. One host mechanism to limit bacterial interactions with the intestinal epithelium is the presence of mucus layers (*Atuma et al., 2001*; *Johansson et al., 2011*). Intriguingly, the single mucus layer in the small intestine is much less adherent than the mucus layers present in the colon (*Johansson et al., 2011*), which might allow occasional direct bacterial interactions with IECs or other cells near the intestinal epithelium. However, soluble components from enteric bacteria may also readily diffuse through mucus layers. In this case, there may be sporadic defects in tight junction proteins that are required to maintain the intestinal epithelium. Tight junction remodeling is essential to maintain intestinal integrity during apoptosis and extrusion of IECs that are regularly shed from the intestinal epithelium (*Williams et al., 2015*). Future studies will be necessary to determine whether defects in epithelial barrier integrity during extrusion events are linked to the local ISG responses that we observe and further delineation of the factors that render specific regions 'responsive' will be of great interest for follow-up studies.

The preceding findings (*Figure 4*) suggested that the homeostatic ISG response in IECs would provide protection against IEC-tropic viruses, such as mRV. Although bacterial-associated, IFN-independent mechanisms of mRV clearance have been reported (*Shi et al., 2019*; *Zhang et al., 2014*), we found that the signature of homeostatic ISGs in ileum tissue included well-characterized antiviral ISGs (*Figures 1–2*). To investigate whether these homeostatic ISGs protect against MRV, we used *Ifnlr1* IEC-cKO mice that lack homeostatic ISGs (*Figures 4 and 6*) rather than using ABX treatment, which introduces pleiotropic effects on rotavirus infection (*Uchiyama et al., 2014*) and dramatically increases transit time through the intestine (*Baldridge et al., 2015*). Using *Ifnlr1* IEC-cKO mice, we found increase in IEC infection by mRV at early stages of infection compared to *Ifnlr1^flox/flox* littermates (*Figure 9* and *Figure 9—figure supplement 1*). However, the protection offered by IEC expression of *Ifnlr1* was lost by the middle and late stages of infection, consistent with the ability of mRV to antagonize induction of IFN responses once infection is established (*Arnold et al., 2013*). Our observations during initiation of infection may provide important context to observations in other studies that report differing capacity for infection-induced IFN-$\lambda$ to protect against mRV infection (*Lin et al.,*

2016; *Pott et al., 2011*). Ultimately, it is clear that prophylactic administration of exogenous IFN-$\lambda$ protects against mRV infection (*Lin et al., 2016*; *Pott et al., 2011*; *Van Winkle et al., 2020*), providing precedent that homeostatic IFN-$\lambda$ would also be protective when induced by bacterial microbiota prior to infection.

Although we found that homeostatic ISGs provide protection during initiation of mRV infection, it remains unclear how these localized ISG pockets impart this protection. Given our findings in *Figure 8* and *Figure 8—figure supplement 1*, we suggest that these localized ISGs may be indicative of locations that are particularly vulnerable to viral infection if ISGs were not present at the time of viral exposure. However, alternative explanations for the protective effects that we observe are also plausible, such as: (i) an inability to detect the full magnitude of ISG expression by imaging, or (ii) an unknown temporal component to homeostatic ISG signaling that may be coincident with durable ISG protein expression. Given the magnitude of signal amplification in RNAscope in situ hybridization and the similarity of *Ifit1* expression in our imaging data to *Ifit1* expression in public single-cell sequencing datasets, we find it unlikely that there is more widespread homeostatic ISG expression below the limit of our detection by imaging. However, we do think an uncharacterized temporal component of homeostatic ISG signaling is plausible, wherein individual villi may be rapidly and transiently expressing homeostatic ISGs in response to sensing of bacterial microbiota. This temporal model may also include a more durable ISG protein response that is not fully concordant with expression of ISG transcripts. In sum, these findings indicate that preexisting, homeostatic ISGs present in *Ifnlr1*-sufficient mice are protective during initiation of mRV infection, but that endogenous IFN-$\lambda$ does not reduce mRV burden in infected cells. These data highlight the possibility that detection of bacterial microbiota in particularly exposed areas may preemptively activate homeostatic ISGs as a form of anticipatory immunity to protect the intestinal epithelium from enteric viruses.

## Materials and methods

### Mice

All mice were bred using the C57BL/6 background and used within the age of 8–12 weeks; C57BL/6J mice (stock #000664) were purchased from Jackson Laboratories (Bar Harbor, ME) and used as WT. Genetically modified mice included *Ifnlr1*$^{-/-}$ and *Ifnlr1*$^{flox/flox}$ (generated from *Ifnlr1*$^{tm1a(EUCOMM)Wtsi}$ as published; *Baldridge et al., 2017*), *Ifnar1*$^{-/-}$ (B6.129.Ifnar1$^{tm1}$), *Trif*$^{-/-}$ (JAX C57BL/6J-*Ticam1*$^{Lps2}$/J, stock #005037), *Myd88*$^{-/-}$ (JAX B6.129P2(SJL)-*Myd88*$^{tm1.1Defr}$/J, stock #009088), *Irf3*$^{-/-}$ (B6.129S/SvEv-Bcl2l12/Irf3$^{tm1Ttg}$), *Irf7*$^{-/-}$ (B6.129P2-Irf7tm1Ttg/TtgRbrc), *Stat1*$^{-/-}$ (B6.129.-*Stat1*$^{tm1Dlv}$), and Villin-cre (B6.Cg-Tg(Vil1-cre)997Gum/J) mice. *Ifnlr1*$^{-/-}$ and *Ifnar1*$^{-/-}$ mice were bred to *Ifnlr1*$^{-/+}$/*Ifnar1*$^{-/+}$ mice to generate littermate *Ifnlr1*$^{-/+}$/*Ifnar1*$^{-/+}$, *Ifnlr1*$^{-/+}$/*Ifnar1*$^{-/-}$, *Ifnlr1*$^{-/-}$/*Ifnar1*$^{-/+}$, *Ifnlr1*$^{-/-}$/*Ifnar1*$^{-/-}$ offspring.

All mice were maintained in specific pathogen-free facilities at Oregon Health & Science University (OHSU) and Washington University in St Louis (WUSTL). Animal protocols were approved by the Institutional Animal Care and Use Committee at OHSU (protocol #IP00000228) and WUSTL (protocol #20190162) in accordance with standards provided in the *Animal Welfare Act*.

For all experiments, mice were allocated into experimental groups based on genotype with equal representation of individual litters and equal sex ratios.

### Mouse treatments

Mice were administered an ad libitum antibiotic cocktail consisting of: 1 g/L ampicillin, 1 g/L metronidazole, 1 g/L neomycin, and 0.5 g/L vancomycin (Sigma, St Louis, MO) in autoclaved $H_2O$ (OHSU) or in 20 mg/mL grape Kool-Aid (Kraft Foods, Northfield, IL) (WUSTL). Sterile $H_2O$ (OHSU) or Kool-Aid (WUSTL) alone was used as a control. Mice were maintained on ABX or control for 2 weeks prior to harvest.

Recombinant IFN-$\lambda$ was provided by Bristol-Myers Squibb (New York City, NY) as a monomeric conjugate comprised of 20 kDa linear PEG attached to the amino-terminus of murine IFN-$\lambda$. Mice were injected intraperitoneally with IFN-$\lambda$ or an equal volume of PBS vehicle as indicated in figure legends at the indicated time prior to analysis.

Mice were stimulated with of 100 µg of the synthetic dsRNA analogue, poly I:C (R&D, #4287) or PBS by intraperitoneal injection in a 200 µL volume, 2 hr prior to harvest.

Twenty-five µg of the bacterial product LPS (Sigma #L4391), flagellin (Invivogen #tlrl-bsfla; from *Bacillus subtilis*) or CpG (Invivogen #tlrl-1585; Class A CpG oligonucleotide), were perorally administered to mice in 25 µL of sterile PBS. Mice were treated on days 15 and 16 of antibiotic treatment or $H_2O$ control prior to harvest on day 17.

For transplantation of fecal material, antibiotic treatment was stopped and mice were fed 25 µL of fecal mixture by pipet for 2 consecutive days. Fecal mixture was prepared by collecting fecal samples from control mice; a single stool pellet was resuspended in 200 µL of sterile PBS, stool was broken apart by pipetting, and large particulate was allowed to settle for several minutes prior to administration.

## Cell isolation

Epithelial fractions were prepared by non-enzymatic dissociation as previously described (*Nice et al., 2016*). Briefly, mouse ileum was opened longitudinally and agitated by shaking in stripping buffer (10% bovine calf serum, 15 mM HEPES, 5 mM EDTA, and 5 mM dithiothreitol in PBS) for 20 min at 37°C. Lamina propria fractions were prepared by enzymatic digestion and dissociation with the Lamina Propria Dissociation Kit and GentleMacs Dissociator (Miltenyi Biotec, Bergisch Gladbach, Germany). Dissociated cells were collected for use in qPCR analysis, flow cytometry, and magnet enrichment.

## Rotavirus infection of mice

Mouse rotavirus (EC strain) was graciously provided by Andrew Gewirtz (Georgia State University). Viral stocks were generated by inoculating 4- to 6-day-old neonatal BALB/c mice and harvesting the entire gastrointestinal tract upon presentation of diarrheal symptoms 4–7 days later. Intestines were freeze-and-thawed, suspended in PBS, and homogenized in a bead beater using 1.0 mm zirconia-silica beads (BioSpec Products). These homogenates were clarified of debris, aliquoted, and stored at –70°C. The 50% shedding dose (SD50) was determined by inoculation of 10-fold serial dilutions in adult C57BL/6J mice. For stool timecourse studies, mice were inoculated by peroral route with 100 SD50 and a single stool pellet was collected daily for viral quantitation by qPCR. For protection studies, mice were inoculated by intragastric gavage with 5000 SD50, and ileum was isolated and mechanically stripped 12- and 24 hr later for quantitation of viral burden by qPCR and flow cytometry.

## RNA isolation, rDNA isolation, qPCR, and analysis

RNA from tissue and stripped IECs was isolated with TRIzol (Life Technologies, Carlsbad, CA) according to the manufacturer's protocol. RNA from magnet-enriched cells was purified by Zymo *Quick*-RNA Viral Kit (Zymo Research, Irvine, CA). The larger of either 1 µg of RNA or 5 µL of RNA were used as a template for cDNA synthesis by the ImProm-II reverse transcriptase system (Promega, Madison, WI) after DNA contamination was removed with the DNAfree kit (Life Technologies). 16S bacterial rDNA was isolated from stool and intestinal contents with a ZymoBIOMICS DNA kit (Zymo Research, Irvine, CA) kit. Quantitative PCR was performed using PerfeCTa qPCR FastMix II (QuantaBio, Beverly, MA) and the absolute quantities of transcript were determined using standard curves composed of gBlocks (IDT) containing target sequences. Absolute copy numbers from tissue samples were normalized to the housekeeping gene, ribosomal protein S29 (*Rps29*). Taqman assays for selected genes were ordered from IDT (Coralville, IA): *Rps29* (Mm.PT.58.21577577), *Ifit1* (Mm.PT.58.32674307), *Oas1a* (Mm.PT.58.30459792), *Mx2* (Mm.PT.58.11386814), *Stat1* (Mm.PT.58.23792152), *Isg15* (Mm.PT.58.41476392.g). Taqman assays for *Ifnl2/3* and *Ifnb1* were designed previously (*Van Winkle et al., 2020*) and consisted of the following primer-probe sequences: *Ifnl2/3* (Primer 1 – GTTCTC-CCAGACCTTCAGG, Primer 2 – CCTGGGACCTGAAGCAG, Probe – CCTTGCAGGCTGAGGTGGC); *Ifnb1* (Primer 1 – CTCCAGCTCCAAGAAAGGAC, Primer 2 – GCCCTGTAGGTGAGGTTGAT, Probe – CAGGAGCTCCTGGAGCAGCTGA). Murine rotavirus was detected using Taqman primer-probe sets specific for 422–521 of GeneBank sequence DQ391187 as previously described (*Fenaux et al., 2006*) with the following sequences: Primer 1 – GTTCGTTGTGCCTCATTCG, Primer 2 – TCGGAACG-TACTTCTGGAC, Probe – AGGAATGCTTCAGCGCTG; and universal bacterial 16S rDNA was detected using Taqman primer-probe sets with previously designed sequences (*Nadkarni et al., 2002*): Primer 1 – GGACTACCAGGGTATCTAATCCTGTT, Primer 2 – TCCTACGGGAGGCAGCAGT, Probe – CGTA TTACCGCGGCTGCTGGCAC.

## RNAscope

Swiss rolls of intestinal tissue were fixed in 10% neutral-buffered formalin for 18–24 hr and paraffin-embedded. Tissue sections (5 μm) were cut and maintained at room temperature with desiccant until staining. RNA in situ hybridization was performed using the RNAscope Multiplex Fluorescent v2 kit (Advanced Cell Diagnostics, Newark, CA) per protocol guidelines. Staining with anti-sense probes for detection of *Ifit1* (ACD, #500071-C2), *Usp18* (ACD, #524651-C1), and MRV (ACD, #1030611-C1) was performed using ACDBio protocols and reagents. MRV probes were designed to target 2–1683 of DQ391187 against NSP3, VP7, and NP4. Slides were stained with DAPI and mounted with ProLong Gold antifade reagent (ThermoFisher), and imaged using a Zeiss ApoTome2 on an Axio Imager, with a Zeiss AxioCam 506 (Zeiss).

Collected images were batch processed in Zeiss Zen 3.1 using unstained control slides to set background values and quantified using ImageJ. Area of *Ifit1* was determined by positive *Ifit1* fluorescent area relative to the total fluorescent area of the tissue section. *Ifit1*+ mRV-infected cells were defined as mRV+ particles with greater than 5 μm area with a maximum *Ifit1* intensity greater than 10% above background. These *Ifit1*+ cells were then divided by the total number mRV+ cells to determine the percentage of *Ifit1*+ mRV-infected cells.

## Flow cytometry and magnet enrichment

Dissociated cells were collected and stained for flow cytometry. Cells were stained with Zombie Aqua viability dye (BioLegend), Fc receptor-blocking antibody (CD16/CD32; BioLegend), anti-EpCAM (clone G8.8; BioLegend), and anti-CD45 (clone 30-F11; BioLegend). For analysis of murine rotavirus infection, cells were stained with anti-rotavirus (polyclonal; ThermoFisher, #PA1-7241) followed by goat anti-rabbit secondary (ThermoFisher). All data were analyzed using FlowJo software (BD Biosciences). Gates were set based on unstained and single-fluorophore stains. IECs were selected by gating on live, EpCAM-positive, CD45-negative cells. Gates for murine rotavirus infection were set based on naïve samples.

Where indicated, dissociated cells were enriched using MojoSort Mouse anti-APC Nanobeads (BioLegend, #480072) after flow cytometry staining for anti-EpCAM and anti-CD45 with APC fluorophores by following manufacturer protocols.

## RNA sequencing and expression analysis

WT C57BL/6J or *Ifnlr1*[-/-] mice were administered ad libitum Kool-aid or ABX for 2 weeks, or WT mice were administered 25 μg recombinant IFN-$\lambda$ for 1 day, then ileal segments lacking Peyer's patches were harvested and RNA-seq was performed as prior (*Park et al., 2016*). mRNA from ilea was purified with oligo-dT beads (Invitrogen, Carlsbad, CA) and cDNA was synthesized using a custom oligo-dT primer containing a barcode and adaptor-linker sequence, degradation of RNA-DNA hybrid following single-strand synthesis, and ligation of a second sequencing linker with T4 ligase (New England Biolabs, Ipswich, MA). These reactions were cleaned up by solid phase reversible immobilization (SPRI), followed by enrichment by PCR and further SPRI to yield strand-specific RNA-seq libraries. Libraries were sequenced with an Illumina HiSeq 2500 with three to four mice were included in each group. Samples were demultiplexed with second mate, reads were aligned with STAR aligner and then counted with HT-Seq. DEGs were identified using DESeq2 (*Love et al., 2014*) based on cutoffs of twofold change, and an inclusive p-value < 0.5. Standard GSEA was performed to identify enrichments in IFN-$\lambda$ response genes. RNA-seq data were uploaded to the European Nucleotide Archive (accession #PRJEB43446).

## 16S rRNA gene illumina sequencing and analysis

For sequencing of the 16S rRNA gene, primer selection and PCRs were performed as described previously (*Caporaso et al., 2011*). Briefly, each sample was amplified in triplicate with Golay-barcoded primers specific for the V4 region (F515/R806), combined, and confirmed by gel electrophoresis. PCRs contained 18.8 μL RNase/DNase-free water, 2.5 μL of 10× High Fidelity PCR Buffer (Invitrogen, 11304–102), 0.5 μL of 10 mM dNTPs, 1 μL 50 of mM MgSO4, 0.5 μL each of forward and reverse primers (10 μM final concentration), 0.1 μL of Platinum High Fidelity Taq (Invitrogen, 11304–102), and 1.0 μL genomic DNA. Reactions were held at 94°C for 2 min to denature the DNA, with amplification proceeding for 26 cycles at 94°C for 15 s, 50°C for 30 s, and 68°C for 30 s; a final extension of 2 min

at 68°C was added to ensure complete amplification. Amplicons were pooled and purified with 0.6× Agencourt AMPure XP beads (Beckman-Coulter, A63882) according to the manufacturer's instructions. The final pooled samples, along with aliquots of the three sequencing primers, were sent to the DNA Sequencing Innovation Lab (Washington University School of Medicine) for sequencing by the 2 × 250 bp protocol with the Illumina MiSeq platform.

Read quality control and the resolution of amplicon sequence variants were performed in R with DADA2 (*Callahan et al., 2016*). Non-bacterial amplicon sequence variants were filtered out. The remaining reads were assigned taxonomy using the Ribosomal Database Project (RDP trainset 16/release 11.5) 16S rRNA gene sequence database (*Cole et al., 2014*). Ecological analyses, such as alpha-diversity (richness, Shannon diversity) and beta-diversity analyses (UniFrac distances), were performed using PhyloSeq and additional R packages (*McMurdie and Holmes, 2013*). 16S sequencing data have been uploaded to the European Nucleotide Archive (accession #PRJEB43446).

## scRNA-seq analyses

A mouse scRNA-seq dataset generated by Haber et al. was accessed from NCBI's Gene Expression Omnibus (GEO), accession #GSE92332. A human pediatric scRNA-seq dataset generated by Elmentaite et al. was accessed in processed form from The Gut Cell Atlas, with raw data also available on EMBL-EBI Array Express, accession #E-MTAB-8901. Files were analyzed in R using Seurat (v. 3.2.2) (*Stuart et al., 2019*). For the Haber et al. dataset, UMI counts were normalized to counts per million, $\log_2$ transformed, and homeostatic ISGs were selected for analysis. Data was collated by previously annotated cell type and proportion of cells expressing each individual ISG was calculated. For Elmentaite et al., data was restricted to healthy controls and then collated by previously annotated cell type to determine the proportion of cells expressing individual ISGs.

## Statistical analyses

Sample size estimation was performed based on historical data. Data were analyzed with Prism software (GraphPad Prism software), with specified tests as noted in the figure legends.

## Acknowledgements

The authors would like to thank the following OHSU core facilities: Integrated Genomics Laboratory, Advanced Light Microscopy Core, and Histopathology Core; and the following WUSTL core facilities: Genome Technology Access Center. TJN was supported by NIH grant R01-AI130055 and by a faculty development award from the Sunlin & Priscilla Chou Foundation. JAV was supported by NIH grants T32-GM071338 and T32-AI007472. MTB was supported by NIH grants R01-AI139314, R01-AI141716, and R01-AI141478, the Pew Biomedical Scholars Program of the Pew Charitable Trusts, and a Children's Discovery Institute of Washington University and St Louis Children's Hospital Interdisciplinary Research Initiative grant (MI-II-2019–790). The funders had no role in study design, data collection, and interpretation, or the decision to submit the work for publication.

## Additional information

### Funding

| Funder | Grant reference number | Author |
| --- | --- | --- |
| National Institutes of Health | R01-AI130055 | Timothy J Nice |
| National Institutes of Health | T32-GM071338 | Jacob A Van Winkle |
| National Institutes of Health | T32-AI007472 | Jacob A Van Winkle |
| National Institutes of Health | R01-AI139314 | Megan T Baldridge |

| Funder | Grant reference number | Author |
|---|---|---|
| National Institutes of Health | R01-AI141716 | Megan T Baldridge |
| National Institutes of Health | R01-AI141478 | Megan T Baldridge |
| Pew Charitable Trusts | | Megan T Baldridge |
| Washington University School of Medicine in St. Louis | MI-II-2019-790 | Megan T Baldridge |

The funders had no role in study design, data collection and interpretation, or the decision to submit the work for publication.

### Author contributions

Jacob A Van Winkle, Conceptualization, Investigation, Methodology, Visualization, Writing - original draft, Writing - review and editing; Stefan T Peterson, Elizabeth A Kennedy, Michael J Wheadon, Harshad Ingle, Chandni Desai, Rachel Rodgers, David A Constant, Austin P Wright, Maxim N Artyomov, Sanghyun Lee, Investigation; Lena Li, Project administration; Megan T Baldridge, Timothy J Nice, Conceptualization, Funding acquisition, Investigation, Methodology, Project administration, Supervision, Writing - review and editing

### Author ORCIDs

Stefan T Peterson ⬥ http://orcid.org/0000-0002-2984-6400
Michael J Wheadon ⬥ http://orcid.org/0000-0002-2579-3786
Austin P Wright ⬥ http://orcid.org/0000-0002-1447-5205
Megan T Baldridge ⬥ http://orcid.org/0000-0002-7030-6131
Timothy J Nice ⬥ http://orcid.org/0000-0002-4471-7666

### Ethics

All mice were maintained in specific-pathogen-free facilities at Oregon Health & Science University (OHSU) and Washington University in St. Louis (WUSTL). Animal protocols were approved by the Institutional Animal Care and Use Committee at OHSU (protocol #IP00000228) and WUSTL (protocol #20190162) in accordance with standards provided in the Animal Welfare Act.

### Decision letter and Author response

Decision letter https://doi.org/10.7554/eLife.74072.sa1
Author response https://doi.org/10.7554/eLife.74072.sa2

## Additional files

### Supplementary files
• Transparent reporting form

### Data availability

RNA-seq data were uploaded to the European Nucleotide Archive under accession #PRJEB43446. Source data files are provided for each figure and contain the numerical data used to generate the figures.

The following dataset was generated:

| Author(s) | Year | Dataset title | Dataset URL | Database and Identifier |
|---|---|---|---|---|
| Baldridge et al | 2021 | A homeostatic interferon lambda response to the bacterial microbiome stimulates preemptive antiviral defense within discrete pockets of intestinal epithelium | http://www.ebi.ac.uk/ena/data/view/PRJEB43446 | EBI, PRJEB43446 |

The following previously published datasets were used:

| Author(s) | Year | Dataset title | Dataset URL | Database and Identifier |
|---|---|---|---|---|
| Haber et al | 2017 | A single-cell survey of the small intestinal epithelium | https://www.ncbi.nlm.nih.gov/geo/query/acc.cgi?acc=GSE92332 | NCBI Gene Expression Omnibus, GSE92332 |
| Elmentaite et al | 2020 | Single-cell sequencing of developing human gut reveals transcriptional links to childhood Crohn's disease | https://www.ebi.ac.uk/ena/browser/view/PRJEB37689 | EBI, PRJEB37689 |

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
