## [Editor Report]

The paper shows that homeostatic interferon-stimulated gene expression in the mouse intestinal epithelium (which is not uniform but concentrated in mature epithelial pockets) depends on the presence of bacterial microbiota and intestinal epithelial cell-intrinsic expression of the IFN-λ receptor associated with leucocyte IFN-λ production. Mouse rotavirus infection (an intestinal epithelial pathogen) is more effectively initiated in the absence of this homeostatic IFN-λ response although it remains unclear how the localized pockets of interferon-stimulated gene expression impart protection.

---

## [Decision Letter]

**Decision letter after peer review:**

[Editors’ note: the authors submitted for reconsideration following the decision after peer review. What follows is the decision letter after the first round of review.]

Thank you for submitting the paper "A homeostatic interferon-λ response to bacterial microbiota stimulates preemptive antiviral defense within discrete pockets of intestinal epithelium" for consideration by *eLife*. Your article has been reviewed by 4 peer reviewers, one of whom is a member of our Board of Reviewing Editors, and the evaluation has been overseen by a Senior Editor. The reviewers have opted to remain anonymous.

We are sorry to say that, after consultation with the reviewers, we have decided that this work cannot be considered in its current form for publication by *eLife*. The reviewers engaged in a careful consultation round before this decision was reached. Although there was substantial interest in the findings, there were concerns over the level of novelty, the underlying mechanisms of the localized interferon response and how these may be influenced by microbiota variation. The *eLife* policy is not to invite a revision if significant further experimental work is considered necessary. Although the reviewers concluded that the work necessary was outside the boundaries of a revision request, we would be interested to receive a further submission as a new paper if you are able to address the concerns.

*Reviewer #1 :*

In this paper the authors provide new information about the relative importance of the type I and type III interferon-driven gene expression and anti-viral responses, particularly focused on the role of the Intestinal microbiota to maintain background levels of type III (interferon λ) signaling. They show with interferon λ administration as a positive control, and antibiotic-mediated microbiota biomass depletion that low background levels of type III interferon-driven gene expression are mediated by the microbiota. Heterozygous mouse strain combinations for epithelial specific either type I or type III interferon receptor deficiency shows that the effect is type III mediated. In-situ hypbridisation shows that type III-driven gene expression is highly discontinuous in the epithelial layer and mainly at the villous tips. Rotavirus infection shows slightly accelerated kinetics in the absence of the type III receptor-signaling. Since intestinal type I and type III interferon responses are well described to occur, this provides a distinction between the two signaling pathways the consequent antiviral responses and the role of the microbiota in maintaining a basal level of type III signaling.

This is a highly mature paper with a consistent set of experiments supporting the role of type III interferon signaling as a consequence of microbiota colonization in the intestine. The strengths of the paper derives from experimental contrasting use of epithelial-specific IFNLR deficiency and IFNAR deficiency, control treatments of interferon λ in verifying gene sets, localization of the homeostatic and IFNL-stimulated response by in situ hybridisation and the use of the functional rotavirus response in the strain combination context.

I consider this is almost publication ready.

Questions and suggestions that I have regarding the data and interpretation are as follows.

1. Do the authors consider that the relative importance of type I and type III signaling depends on the age of the mice.

2. What happens to the localization of the type III signal as control stimulation or viral infection proceeds? Presumably the time trajectory of in situs in these different contexts has not been done.

3. Do microbiota composition, bacteriophage lytic phases, or diet influence the homeostatic response.

4. I think that the authors should consider citing the basal signaling in Caco2 cells and organoids (Hakim Sci. Rep. 2018 8:8341) and the interferon-independent (Shi 2019, Cell 179, 644-658) papers in the discussion.

*Reviewer #2:*

In this work, Van Winkle et al. examine contributions of interferons and microbiota to innate responses in the intestine, and which cell types are involved. Previous work by this team and others demonstrated that microbiota influence interferon responses in the intestine, which can affect infection with several enteric viruses, either increasing infection or decreasing infection depending on the viral system. Here, the team profiles expression of ISGs, examines which cells produce IFNs and express ISGs, whether these responses are microbiota dependent, and examines the effect on infection with murine rotavirus.

Strengths:

The most interesting aspect of this study is the observation that ISG expression in the intestine is extremely patchy and limited to a few mature enterocytes. Figure 6 is stunning, and these data are supported by beautiful controls (robust/broad ISG expression everywhere in the intestine of mice treated IP with IFNlambda in Figure S4, but loss of expression in mice treated with ABX, and confirmation in public single cell RNA-seq data sets). This is the most unique and significant contribution of the study.

Other aspects of the study are also well done with appropriate controls (multiple panels with ISG levels throughout, etc.). The team uses a variety of mouse strains, treatments, etc. to support their claims.

Weaknesses:

The primary weakness lies in significance, partially based on past work. The observation about patchy ISG expression in mature enterocytes is very cool, but it remains unknown why/how this happens and whether there are any functional consequences. This reviewer understands that this is a very tough problem and may take time to figure out. Additionally, much work has already been done with microbiota and IFNlambda effects on enteric virus infection, making many of the findings here overlapping or redundant with prior work: IFNlambda effects on IECs and viral infection (Sommereyns Plos Path 2008, Baldridge JVI 2017), leukocytes as the source of IFNlambda (Mahlakoiv Plos Path 2015), and microbiota-mediated innate immune modulation and effects of IFNlambda (e.g., Baldridge Science 2015, Nice Science 2015).

For Figure 5 and S3 data, the authors examine gene expression in intestinal cells collected from mice and then enriched for epithelial vs. lamina propria cell populations. They see significantly reduced expression of IFN upon ABX treatment in just one case-harvested epithelial cells that have been enriched for CD45+ cells (leukocytes) have reduced IFNlambda expression. Lamina propria CD45+ cells do not show significantly reduced IFNlambda. This is surprising. While there are low levels of CD45+ cells in epithelial cell preps (as they show in Figure S3), there are higher levels of CD45+ cells in the LP preps. It seems odd that the only significant difference lies with these immune cells in the epithelial cell prep, and this effect is not observed in LP preps, which should contain many more CD45+ cells. Given the authors' conclusion that leukocytes are the source of IFNlambda, this deserves more follow-up or at least some discussion. Is there something special about CD45+ cells from epithelial preps? Is it possible these are contaminating cells? Or is it simply an issue with spread in the data?

In Figures 8 and S6, the authors examine Ifit1 expression in WT or mutant mice with different treatments (ABX, LPS, fecal transfer), to determine if LPS or fecal transfer can restore gene expression. Due to low expression levels and a high degree of spread in the data, they resort to binning the primary gene expression data (8H and S6A; ABX vs. ABX/LPS not significantly different) into + vs. – categories (Figure 8J, S6E; ABX vs. ABX/LPS now are significantly different) to allow different statistical analysis. From this, they conclude that ISG expression is partially restored by LPS treatment in ABX-treated mice. Since the data are just around the detection limit and it's a high bar to restore gene expression, it's perhaps understandable that the data are binary, but this isn't the most convincing data set.

*Reviewer #3:*

The authors convincingly show by RNAseq, that microbiota ablation by antibiotics treatment deprives WT mice of a tonic and interferon λ induced interferon stimulated gene (ISG) response. Alongside they test interferon λ receptor ko mice, which do not show a tonic induction of ISGs or a response to antibiotics dependent removal of microbiota. Induction of ISGs in intestinal tissue depends on TLR signaling, which would support a microbiota dependent induction. Intriguingly this ISG response is highly localised in very focussed areas of the intestine. Fecal transfer or LPS treatment restore the antibiotics induced phenotype of basal ISG expression loss and partially restore antiviral protection in intestinal epithelial cells.

These data complement recent publications [PMID: 32380006][PMID: 31269444] of tonic type I interferon signaling induced by microbiota could explain a series of publications showing the importance of microbiota for antiviral defence in the gut.

The authors build a line of arguments based on the correlation of data from IFNLR -/- mice and ABX treated mice. They omit however an important alternative explanation, which might be qualitative differences in the microbiota composition between IFNLR -/- and WT mice.

An explanation for the focussed ISG response in the intestine and how this explains the reduced resistance to enteric viruses is not provided.

An alternative explanation for the observed ISG expression phenotype could be a difference in baseline microbiota composition between WT and IFNLR -/- mice. This should be checked by 16S rRNA gene sequencing of small and large intestinal samples and could be complemented by a fecal transfer from IFNLR-/- mice into ABX treated WT mice followed by measurement of baseline ISGs in the intestine.

Qualitative differences in colonisation could also explain the binary response of WT mice to fecal transfer.

The focussed ISG response shown in Figure 8 is intriguing but difficult to reconcile with the 20x reduced viral titres in WT mice. Does the virus only replicate in these sites? And are commensal microbiota specifically dense in these areas. A multicolour FISH approach could answer both of these points.

*Reviewer #4:*

In this paper from Van Winkle et al., the authors determined that murine intestinal epithelial cells produce several homeostatic ISGs. These homeostatic ISGs are located in the mature enterocytes and are stimulated by the presence of bacterial microbiota. This is an exciting concept as it becomes clearer that all cells in a tissue are not the same. The authors nicely demonstrate that these homeostatic ISGs are only present when the commensal microbiota is also present and that they depend on IFNl signaling.

A main weakness of the paper is that the authors use IFNl treatment to stimulate ISGs and then use this as a basis to determine homeostatic gene levels. However, it has been shown that stem cells have basal ISGs that are IFN independent (Wu…Rice, Cell 2018), which means that the only cells that could respond to IFN would be the mature cells in the villi. Additionally, as the bacteria would normally only be in contact with the enterocytes and not the crypts then one would not expect to find bacterial induced homeostatic ISGs lower in the crypt-villi axis. Given these, the authors findings while interesting are expected due to the experimental set-up. While several experiments contain complete controls to make all interpretations, several experiments lack controls and many conclusions suffer from over statements, lack of convincing imaging and correlations between infections and ISGs.

– In Figure 1 the authors claim that there is no significant difference between ifnlr-/- +/- ABX. However, this is hard to say from the presented data. There are no clear differences in the Enrichment plots D-F and the values given in B seem to be wrong as it seems very unlikely that both the FDR and the Nominal p-value are the exact same number. Without the correct numbers it is hard to judge the conclusion of this figure.

– On Figure 1, the authors claim that WT mice separate from ifnlr-/- mice, however in Figure 1H, one of the WT mice and ifnlr-/- mice cluster together. The authors should add more mice to show this more clearly or tone down this statement.

– On Figure S2, the IFNL without ABX control is missing to be able to interpret the data set.

– In Figure 3, why are the het samples shown as stripped epithelium and not full tissue as the others in the experiment. It would be good to have all tissue samples or all stripped epithelium to be able to make a full comparison between the data sets.

– Additionally, in Figure 3 the ifr7-/- effect on ISGs levels is very modest and is further decreased with ABX (which is normally considered a criteria for it to not be key for this response) the authors should provide more evidence that irf7 is involved or tone down their conclusion in this regard.

– The impact of ABX and ifnlr-/- seems to be less apparent in the colon than the ileum (e.g. Figure 2A vs B). This is curious since the colon will have the higher bacterial load. Can the authors comment on this?

– Figure 5B, it is curious that there is no induction of ifnb1 in the epithelial fraction upon poly IC treatment. While several studies have shown that the cells are not responsive to type I IFNs, there is no data showing that they cannot produce type I IFNs upon MAMP stimulation. Can the authors explain this?

– Figure S3, why are the values of CD45+ cells not closer to 100% post-enrichment? If there are still 50% of non-CD45+ cells that are being analyzed, what are they and can they impact the results shown in Figure 5?

– Figure S4, high magnification images are needed to see if the crypts are able to respond to IFNs in a murine model to clearly demonstrate whether it would be possible for them to have IFN dependent homeostatic ISGs or whether the authors are already selecting for a phenotype only present in mature enterocytes.

– Figure S5, the colon data is not very convincing as the ifnlrKO shows some background staining which does not look much different from the other images. Additionally, it is not clear why only floxed and dIEC mice are used in this figure and there are no WT mice used as in Figure 6. It would be good to have the matching animals to better compare the data. It would also be important to include high mag images to clearly see if the ISG distribution is the same (as it does not seem to be the same tip distribution that is seen in the ileum). This figure should also include a quantification as provided in other figures.

– Figure 7, crypt cells are not a cell type. There are stem cells, TA and other progenitors present that make up the crypt. This should be renamed to correctly indicate the type of cells in this population.

– Additionally, the scRNA-Seq used for human analysis was a bit strange. This paper used mainly embryos and children to derive the cell types. The fact that only a few cell types were highly present seems to indicate that maybe this was not the best paper. It would be good to compare to stronger papers such as Wang et al. JEM 2020, Fuji et al. Cell Stem Cell 2018. There is also a paper using human ileum showing that stem cells and enterocytes each have their own basal ISGs which are stem cell specific or enterocytes specific (Triana, 2020). The statements about the human data should be toned down or a more robust scRNA data set should be used.

– Figure S6D, why are there different numbers of mice used for each ISG? If the RNA is harvested, then the same RNA can be used to assess all three ISGs. It seems strange that of 30 mice tested for ifit1, only 17 were tested for stat1. These missing numbers seems strange and should be included.

– The homeostatic ISGs are only found in the tip of the villi, and rotavirus infection normally occurs at the tips of the villi, the authors claim that these homeostatic ISGs are more protective against infection. To make this claim the authors need to perform a co-staining of rotavirus infected tissue to show that the villi that are infected are not the villi which express the homeostatic ISGs. In general, rotavirus staining is needed as the FACS data suggests that a very low numbers of cells are infected which makes it hard to justify the authors claims.

– The authors claim that IFN is being produced from CD45+ epithelium derived lymphocytes. To support and strengthen this claim, it would be important to perform RNA scope with markers for these cells to determine if the sparse ISG induction seen in the tissue is correlates with the localization of these cells.

– Since the authors claim is that the homeostatic ISGs come from sensing of bacterial components, and that LPS can rescue this phenotype, then the authors should determine if tlr4-/- mice also lack the presence of homeostatic ISGs.

– Furthermore, the authors claim in the discussion that microbiota components are responsible for the homeostatic levels and that they signal through several MAMPs. However, this is not shown in the paper. In the manuscript the authors only tested LPS. They also used CpG which will signal through MyD88 but does not induce this response. To make the claim that multiple MAMPs can be used, then several other MAMP agonists should be used to show that this is not only a LPS phenotype (e.g. flagellin).

[Editors’ note: further revisions were suggested prior to acceptance, as described below.]

Thank you for sending your article entitled "A homeostatic interferon-λ response to bacterial microbiota stimulates preemptive antiviral defense within discrete pockets of intestinal epithelium" for peer review at *eLife*. Your article is being evaluated by 3 peer reviewers, one of whom is a member of our Board of Reviewing Editors, and the evaluation is being overseen by Carla Rothlin as the Senior Editor.

*Reviewer #1:*

The authors have completed a careful revision of the manuscript in accordance with the original critiques.

Apart from the duplication of the Figure 3 as Figure 2 which is presumably a submission error, I still have some concerns.

The antibiotic treatments in combination with the strain combinations leave overlapping ranges of interferon-stimulated genes (ISGs) that gives some uncertainty of interpretation (for example in the Ifnlr1-deficient for Stat1 transcripts which is non-significant with respect to wild-type for the left panel of 4A). Presumably the noise in the system also makes the Irf7-deficient result challenging to interpret in the left panel of 3B.

The patchy expression of the ISGs in the histological panels is clear, but I am worried about the security of the interpretations of LPS or flagellin recovery of the effects through stratification of those animals where the effect is or is not seen when there may be as few as 4 signals in the positive control (Figure 8A).

Finally, the rotavirus 'preemptive' protection is at best partial and likely context (for example microbiota composition) dependent, so it would be a surprise to me if all investigators trying to repeat this obtained exactly the same result. I think that the authors would have to be more cautious in their conclusions and in their abstract to make the paper publishable.

*Reviewer #2:*

The authors convincingly excluded that qualitative differences in the microbiota between WT and IFNLR ko mice are responsible for the observed differences in baseline ISG expression.

The data in Figure 9I are however not convincingly showing that RV preferentially replicates in IFIT1 negative cells. The statement "we generally found that mRV RNA was not present in villi that have IFIT1 expression" in line 603 is not supported by the provided images. If there is a biological effect, the authors would have to substantiate it by providing a quantification of relative number of RV positive cells in presence or absence of IFIT1 co-staining.

This is problematic since the authors central claim is the increased defence against viral pathogens as a direct consequence of microbiota induced IFNL The current dat a would suggest a rather indirect mechanism.

*Reviewer #3:*

The revisions to the manuscript have improved the clarity and strengthened several concerns. I believe the manuscript is close to be ready for publication however there are still a few questions that could use a bit more quantifications to improve their strength.

Figure 8 and S9. While I appreciate the difficulty in these experiments by adding back the fecal transplant or adding the TLR agonists. I am still concerned about the strength of the data and claiming that they can partially restore the phenotype.

Figure 9- Thanks for including the co-stainings, this helps to see how virus infection is relative to the IFIT1 stainings, however these images are lacking quantification. Additionally, it would be important to plot the % of co-localization of IFIT and rotavirus as it seems that in several examples there is more overlap than suggested in the text.

Figure 10 – The 96h time was performed only 1 time. It would be important to have this repeated more than one time and to have the colocalization of IFIT and rotavirus for this time point as well. As Figure 9 indicates that there is a lot of virus shedding still taking place at this time, it would be important to clarify how this correlates with the ISG expression.

---

## [Author Response]

[Editors’ note: the authors resubmitted a revised version of the paper for consideration. What follows is the authors’ response to the first round of review.]

Reviewer #1:[…]Questions and suggestions that I have regarding the data and interpretation are as follows.1. Do the authors consider that the relative importance of type I and type III signaling depends on the age of the mice.

This is an important point, and we are aware of studies that indicate age of mice can influence IEC responsiveness to Type I IFN. To emphasize these findings, we have updated our introduction to highlight studies that explored this topic on (Lines 54-56): “The responsiveness of IECs to type I IFN appears to be developmentally regulated because the IECs of adult mice exhibit weaker type I responses than IECs of neonatal mice (Lin et al., 2016).”

2. What happens to the localization of the type III signal as control stimulation or viral infection proceeds? Presumably the time trajectory of in situs in these different contexts has not been done.

We find the temporal aspect of this localized homeostatic type III signaling to be incredibly interesting. We are unable to perform time trajectories of individual mice by in situ hybridization due to the terminal nature of this approach. However, we have now included ileal *Ifit1* in situ hybridization imaging of representative mice at 12hr, 24hr, and 96hr postinoculation with murine rotavirus. These data are presented in Figure S10, are described in Lines 603-605: “Additionally, we found mRV infection did not dramatically increase area of *Ifit1* expression in the intestine at 24 hours post-inoculation. However, at 96 hours post-inoculation, we found a significant increase in epithelial *Ifit1* expression in the ileum relative to early times post-inoculation.”

3. Do microbiota composition, bacteriophage lytic phases, or diet influence the homeostatic response.

To address this important point, we now provide evidence that microbiota composition does not influence the homeostatic ISG response detailed in this manuscript. We find no difference in the number of observed bacterial species, Shannon Diversity Index, or UniFrac distance when we performed 16S rRNA sequencing on stool from *Ifnlr1* +/+, *Ifnlr1* +/, and *Ifnlr1 -/-* mice. We have included these data in Figure S1 and described them in text (Lines 117-120): “To rule out contributions of Ifnlr1 toward an altered intestinal bacterial microbiota, we performed 16S rRNA sequencing on stool from *Ifnlr1^+/+^, Ifnlr1^+/-^, and Ifnlr1^-/-^* mice and did not find differences in α-diversity and β-diversity measurements.”

Secondly, we find the possibility of bacteriophage mechanisms to be intriguing, but have collected no data on this topic. We briefly collaborated with Dr. Brooke Napier’s group at Portland State University to assess whether ketogenic diet affects homeostatic ISG expression by altering LPS tolerance, but did not observe any interactions in our limited studies. For the reviewer’s benefit, in Author response image 1 we have data from stripped intestinal epithelium of mice fed standard chow and a keto diet.

**Author response image 1. sa2fig1:** 

4. I think that the authors should consider citing the basal signaling in Caco2 cells and organoids (Hakim Sci. Rep. 2018 8:8341) and the interferon-independent (Shi 2019, Cell 179, 644-658) papers in the discussion.

We agree that basal interferon responses and interferon-independent bacterial clearance of murine rotavirus is very interesting. We have expanded the discussion to include citations to both references.

We mention Hakim et al. in lines 659-661: “ Although there are previous reports of basal, non-receptor-dependent ISG expression in immortalized and primary cell lines (Hakim et al., 2018), the homeostatic response that we report here is dependent on the IFN-λ receptor.” and Shi et al. in lines 726-729: “Although bacterial-associated, IFN-independent mechanisms of mRV clearance have been reported (Shi et al., 2019; Zhang et al., 2014), we found that the signature of homeostatic ISGs in ileum tissue included well-characterized antiviral ISGs (Figures 1-2).”

Reviewer #2:[…]For Figure 5 and S3 data, the authors examine gene expression in intestinal cells collected from mice and then enriched for epithelial vs. lamina propria cell populations. They see significantly reduced expression of IFN upon ABX treatment in just one case-harvested epithelial cells that have been enriched for CD45+ cells (leukocytes) have reduced IFNlambda expression. Lamina propria CD45+ cells do not show significantly reduced IFNlambda. This is surprising. While there are low levels of CD45+ cells in epithelial cell preps (as they show in Figure S3), there are higher levels of CD45+ cells in the LP preps. It seems odd that the only significant difference lies with these immune cells in the epithelial cell prep, and this effect is not observed in LP preps, which should contain many more CD45+ cells. Given the authors' conclusion that leukocytes are the source of IFNlambda, this deserves more follow-up or at least some discussion. Is there something special about CD45+ cells from epithelial preps? Is it possible these are contaminating cells? Or is it simply an issue with spread in the data?

Previous work by Mahlakoiv et al. (Figure S3 of Mahlakoiv 2015) showed that CD45+ cells in intestinal epithelial cell preps express IFN-λ transcripts at homeostasis and therefore, we do not find it surprising to detect IFN-λ transcripts in this cellular fraction. The data depicted in Figure 5 and Figure S6 (previously S3) build upon the findings of Mahlakoiv et al. by confirming that these homeostatic IFN-λ transcripts are dependent upon the presence of bacterial microbiota. Although CD45+ cells are more abundant in LP preps, our and Mahlakoiv et al’s findings suggest that the proximity of CD45+ cells to bacterial stimuli in the intestinal epithelial layer is more indicative of IFN-λ expression than the relative abundance of CD45+ cells. To clarify these points, we have updated the Results section to elaborate on the findings of Mahlakoiv et al. (lines 336-338): “A previous study (Mahlakõiv et al., 2015) noted the presence of IFN-λ transcripts (*Ifnl2/3*) at homeostasis in CD45+ cells within the stripped intestinal epithelium, but not in the lamina propria.”

We additionally emphasize the potential role of CD45+ cells from the lamina propria in our conclusion of Figure 5 (lines 348-350): “From these data, we conclude that epithelium-associated CD45+ leukocytes are the likely source of homeostatic IFN-λ in response to bacterial microbiota, but we do not rule out additional involvement of CD45+ cells in the lamina propria.”

We have also expanded our discussion to include further rationale for CD45+ cell surveillance of the intestinal epithelium for stimuli from the bacterial microbiota (lines 685-690): “Although the cell type that produces homeostatic IFN-λ is unknown, enrichment for *Ifnl2/3* in CD45+ cells within the intestinal epithelial fraction suggests that proximity to bacterial stimuli may be a primary determinant in this response. We suggest that these cells may be actively surveying the intestinal epithelium to detect bacterial MAMPs. The specific CD45+ cell types responsible for producing homeostatic IFN-λ will be a topic of interest for future studies.”

Our future work beyond the scope of this study aims to definitively characterize the IFN-λ source cell type and its requirement for homeostatic ISG expression.

In Figures 8 and S6, the authors examine Ifit1 expression in WT or mutant mice with different treatments (ABX, LPS, fecal transfer), to determine if LPS or fecal transfer can restore gene expression. Due to low expression levels and a high degree of spread in the data, they resort to binning the primary gene expression data (8H and S6A; ABX vs. ABX/LPS not significantly different) into + vs. – categories (Figure 8J, S6E; ABX vs. ABX/LPS now are significantly different) to allow different statistical analysis. From this, they conclude that ISG expression is partially restored by LPS treatment in ABX-treated mice. Since the data are just around the detection limit and it's a high bar to restore gene expression, it's perhaps understandable that the data are binary, but this isn't the most convincing data set.

We appreciate that Reviewer #2 recognizes the difficulties associated with analyzing the datasets present in Figure 8 and Figure S9 (formerly S6). Because control reconstitution with fecal transfer results in incomplete restoration of homeostatic ISGs, we find it compelling that restoration of *Ifit1* signal by LPS in ABX-treated mice is also localized. Although these results are variable, we think that the use of three candidate ISGs and two experimental approaches increases the weight of their evidence. Additionally, we now include experiments testing a role for a distinct MAMP (flagellin) in restoration of homeostatic ISGs in Figure S9. Flagellin results in partial restitution, similar to LPS, further increasing the collective strength of these data.

Reviewer #3:[…]An alternative explanation for the observed ISG expression phenotype could be a difference in baseline microbiota composition between WT and IFNLR -/- mice. This should be checked by 16S rRNA gene sequencing of small and large intestinal samples and could be complemented by a fecal transfer from IFNLR-/- mice into ABX treated WT mice followed by measurement of baseline ISGs in the intestine.

We now provide evidence that microbiota composition does not influence the homeostatic ISG response detailed in this manuscript. We have found no difference in the number of observed bacterial species, Shannon Diversity Index, or UniFrac distance when we performed 16S rRNA sequencing on stool from *Ifnlr1* +/+, *Ifnlr1* +/-, and *Ifnlr1 -/-* mice. We have included these data in Figure S1 and described them in text (Lines 117-120): “To rule out contributions of Ifnlr1 toward an altered intestinal bacterial microbiota, we performed 16S rRNA sequencing on stool from *Ifnlr1^+/+^, Ifnlr1^+/-^, and Ifnlr1^-/-^* mice and did not find differences in α-diversity and β-diversity measurements.”

Furthermore, although early figures used separately bred WT and *Ifnlr1-/-* mouse lines, we went on to validate findings using littermate *Ifnlr1^flox/flox^* and *Ifnlr1*^ΔIEC^ mice in subsequent figures (Figures 4 and 6). These littermate comparisons in combination with 16S rRNA sequencing suggest that unique microbiota composition is not the primary determinant of the homeostatic IFN-λ response.

Qualitative differences in colonisation could also explain the binary response of WT mice to fecal transfer.

It is plausible that localized differences in colonization by specific members of the bacterial microbiota may exist and may explain the binary response of WT mice to fecal transfer. However, because administration of free LPS (and also flagellin in revised manuscript) to ABX-treated mice results in binary restoration of *Ifit1* signal, our model is that bacterial colonization is not necessary for localization of ISG signal. However, we have not fully ruled out a contributory role for qualitative differences in bacterial colonization, so we now include discussion of this possibility into the discussion in Lines 698-701: “The basis for this pattern is unknown; however, it may reflect the distribution of cells capable of sensing bacterial microbiota, distinct microenvironments within the gastrointestinal tract, or qualitative differences in bacterial colonization.”

The focussed ISG response shown in Figure 8 is intriguing but difficult to reconcile with the 20x reduced viral titres in WT mice. Does the virus only replicate in these sites? And are commensal microbiota specifically dense in these areas. A multicolour FISH approach could answer both of these points.

We agree that experiments to visualize sites of virus replication are important context. Therefore, we have now included multicolor FISH depicting the distribution of *Ifit1* and murine rotavirus transcript at 24 hours post-infection (Figure 9I) and have included description of these results in-text (Lines 598-602): “To further contextualize the role of homeostatic ISGs in protection against mRV, we performed in situ hybridization for both mRV and *Ifit1* in the ileum of WT mice at 24 hours post-inoculation (Figure 9I). We observed variability in the magnitude of mRV infection between mice at this timepoint; however, we generally found that mRV RNA was not present in villi that have *Ifit1* expression.” To summarize, we observe that mRV-infected IECs are primarily *Ifit1* negative, and *Ifit1*-positive villi are rarely infected by mRV. These observations are consistent with mRV evasion of ISG production by infected cells, and imply (but do not prove) a protective role for preexisting ISG expression. We propose that regions with homeostatic ISG expression may have increased chance of initial encounter with mRV and are resistant to infection, but that these regions are not exclusive sites of mRV encounter. In addition to imaging, this model is partly informed by our observations that luminal MAMP administration results in localized ISG response and suggests differential access of luminal contents (Figure 8, Figure S9).

Local variation in bacterial microbiota concentration is an attractive hypothesis for localized ISG expression, but we have been unable to collect sufficient evidence to support it. Our attempts to retain mucus and bacteria within our tissue sections were not consistently successful across the large tissue sections needed to properly survey for localized ISG expression. Additionally, as mentioned above, administration of luminal MAMP to ABX-treated mice partially restores *Ifit1* signal, so we infer that qualitative differences in bacterial colonization are insufficient to explain the localization of ISG signal.

Reviewer #4:[…]– In Figure 1 the authors claim that there is no significant difference between ifnlr-/- +/- ABX. However, this is hard to say from the presented data. There are no clear differences in the Enrichment plots D-F and the values given in B seem to be wrong as it seems very unlikely that both the FDR and the Nominal p-value are the exact same number. Without the correct numbers it is hard to judge the conclusion of this figure.

In response to these and other reviewer comments, we have made multiple changes listed below to more clearly present our GSEA analysis. Together, our conclusions are unchanged from our prior submission, but the presentation is improved in clarity and includes additional information about gene set changes in our treatment groups.

1) We have removed genes with 0 counts in all samples because they do not provide information for enrichment comparisons and decreased visual clarity of the prior enrichment plots.

2) To improve upon our prior “circular” approach of using IFN-λ treatment response (Lee et al.) to test for ISG enrichment, our revised manuscript analyzes our data by GSEA for enrichment of the independently-curated HALLMARK gene sets, which include ISG sets and other pathways relevant to intestinal homeostasis (Figure S2) (Liberzon et al., 2016).

3) We now report only FDR, which is adjusted for multiple comparison testing such as the hallmark gene sets. The values in the previous Figure 1B (both FDR and p-value) were equal because of the usage of a single gene set, where FDR takes into account multiple hypotheses testing with multiple gene sets.

4) In our new analysis using HALLMARK gene sets (Figure S2), we find that INTERFERON_Α_RESPONSE genes are most upregulated in the ileum with IFN-λ stimulation, consistent with the known overlap between type I and III IFN signaling pathways. Therefore, we use this independently-curated set of ISGs for enrichment plots in Figure 1B-E.

– On Figure 1, the authors claim that WT mice separate from ifnlr-/- mice, however in Figure 1H, one of the WT mice and ifnlr-/- mice cluster together. The authors should add more mice to show this more clearly or tone down this statement.

We agree that our prior use of hierarchical clustering to support the claim that there is no significant difference between *Ifnlr1-/-* mice regardless of ABX-treatment was not convincing. Therefore, we have instead included statistical analysis of count values of individual homeostatic ISGs in Figure 1G. We find there is no statistically significant difference between the ISGs of ABX-treated *Ifnlr1-/-* mice and conventional *Ifnlr1-/-* mice. These data provide additional quantification that augments our initial gene set enrichment analysis. In light of our new analysis, we have amended our commentary on the results of the heatmap in Figure 1G (formerly Figure 1H) on lines 154-162: “Comparison of homeostatic ISG transcript counts between experimental treatments revealed similar insights as prior GSEA analysis. WT mice treated with IFN-λ had higher expression of all homeostatic ISGs than untreated mice, whereas WT mice with a conventional microbiota had higher expression of homeostatic ISGs than *Ifnlr1^-/-^* mice and ABX-treated mice of both genotypes (Figure 1G). We did not detect additional decreases in these homeostatic ISGs in ABX-treated *Ifnlr1^-/-^* mice relative to conventional *Ifnlr1^-/-^* mice, suggesting that *Ifnlr1* is necessary for expression of homeostatic ISGs (Figure 1G). These results indicate that there is modest but significant expression of ISGs at homeostasis that is lost with *Ifnlr1* deficiency or ABX treatment.”

– On Figure S2, the IFNL without ABX control is missing to be able to interpret the data set.

We do not think that the omission of IFN-λ-treated control mice precludes interpretation of this dataset. We have reported these supplementary data in lines 217-220 where we write, “Stimulation with small amounts of IFN-λ rescued ISG expression in whole tissue (Figure S5, formerly Figure S2), indicating that reduction of homeostatic ISG expression upon treatment with ABX is not due to an inability of the intestine to respond to IFN-λ”.

Our data are sufficient to show that IFN-λ injection of ABX-treated mice can significantly increase three candidate ISGs in the ileum, which suggests that the effect of ABX is not on the responsiveness of the intestine to IFN-λ, but rather, the production of endogenous IFN-λ.

– In Figure 3, why are the het samples shown as stripped epithelium and not full tissue as the others in the experiment. It would be good to have all tissue samples or all stripped epithelium to be able to make a full comparison between the data sets.

We have updated Figure 3D to display analysis of full tissue rather than stripped epithelium in the interest of consistency with data in Figure 3A-C.

– Additionally, in Figure 3 the ifr7-/- effect on ISGs levels is very modest and is further decreased with ABX (which is normally considered a criteria for it to not be key for this response) the authors should provide more evidence that irf7 is involved or tone down their conclusion in this regard.

We have toned down our conclusion when describing Figure 3B (Lines 273-279): “However, we observed a modest (twofold) decrease in Ifit1 expression in both the ileum and colon of *Irf7^-/-^* mice when compared to WT mice (Figure 3B). Although IRF7 is implicated by these data, it does not appear to be strictly required for homeostatic expression of *Ifit1* because expression is further reduced by ABX treatment (Figure 3B). These data suggest that IRF7 is not necessary for the homeostatic ISG response to the bacterial microbiota.”

– The impact of ABX and ifnlr-/- seems to be less apparent in the colon than the ileum (e.g. Figure 2A vs B). This is curious since the colon will have the higher bacterial load. Can the authors comment on this?

We observe that homeostatic ISGs are expressed at lower levels in colonic tissue as compared to ileal tissue; therefore, the decreases of these ISGs with ABX and in *Ifnlr1-/-* mice are less robust than their analogous condition in ileal tissue. We have now included paired comparisons of *Ifit1* expression from the ileum and colon of WT mice in Figure S4, and thank

the reviewer for drawing our attention to clarify this point

– Figure 5B, it is curious that there is no induction of ifnb1 in the epithelial fraction upon poly IC treatment. While several studies have shown that the cells are not responsive to type I IFNs, there is no data showing that they cannot produce type I IFNs upon MAMP stimulation. Can the authors explain this?

The lack of *Ifnb1* induction by IECs in the epithelial fraction upon poly I:C treatment is consistent with a prior publication by Mahlakoiv et al. (Mahlakoiv 2015, Figure 2B), which reported that the bulk epithelial fraction lacks *Ifnb1* induction upon poly I:C injection. Epithelial cells make up the majority of the epithelial cell fraction; therefore, we expected enriched EpCAM+ cells to lack *Ifnb1* induction, consistent with Mahlakoiv et al.

– Figure S3, why are the values of CD45+ cells not closer to 100% post-enrichment? If there are still 50% of non-CD45+ cells that are being analyzed, what are they and can they impact the results shown in Figure 5?

The initial abundance of CD45+ cells in the intestinal epithelial strip is relatively low (~5%), so our enrichment method does not yield near 100% puruty. Upon purity analysis by flow cytometry, we have noted an abundance of dead EpCAM+ cells. In the interest of transparency regarding the cellular material used for qPCR, we have included all live and dead flow events for our purity analysis (Figure S6), and this point is now further clarified in the revised figure legend. We would suggest that the inclusion of CD45-negative events in the CD45-enriched population does not lead to erroneous conclusions: we find the purity of EpCAM+ cells is quite high (> 90%) post-enrichment, but we cannot detect IFN-λ transcripts in these cells; therefore, the contamination of EpCAM+ events in the CD45-enriched populations does not confound our conclusions from Figure 5.

– Figure S4, high magnification images are needed to see if the crypts are able to respond to IFNs in a murine model to clearly demonstrate whether it would be possible for them to have IFN dependent homeostatic ISGs or whether the authors are already selecting for a phenotype only present in mature enterocytes.

We have updated Figure S7 (formerly figure S4) and now show increased magnification fields as insets. We have also clarified the language in the Results section of Figure 6 to reflect that this stimulation depicts minimal *Ifit1* expression in intestinal crypts, consistent with the lack of homeostatic ISG expression by crypt-associated cells in Figure 6A and, subsequently, Figure 7 (Lines 390-395): “ We determined that localized ISG expression within individual villi was not due to a localized ability to respond to IFN-λ because stimulation with exogenous IFN-λ resulted in *Ifit1* expression within all intestinal villi, but not intestinal crypts (Figure S7). The minimal expression of *Ifit1* within intestinal crypts following exogenous IFN-λ treatment suggests that homeostatic ISGs are localized to mature enterocytes because intestinal crypts do not exhibit robust responses to IFN-λ.”

– Figure S5, the colon data is not very convincing as the ifnlrKO shows some background staining which does not look much different from the other images. Additionally, it is not clear why only floxed and dIEC mice are used in this figure and there are no WT mice used as in Figure 6. It would be good to have the matching animals to better compare the data. It would also be important to include high mag images to clearly see if the ISG distribution is the same (as it does not seem to be the same tip distribution that is seen in the ileum). This figure should also include a quantification as provided in other figures.

In Figure S8 (formerly figure S5), we provided images of *Ifit1* expression in the colon of *Ifnlr1*^Het^/*Ifnar1*^Het^ and littermate mice singly deficient in *Ifnlr1* (*Ifnlr1*^KO^/*Ifnar1*^Het^) or *Ifnar1* (*Ifnlr1*^Het^/*Ifnar1*^KO^). Our manuscript thus includes images from both full and conditional KO mice as well as relevant controls. We are confused regarding the reviewer’s comments about *Ifnlr1^flox/flox^* and *Ifnlr1*^ΔIEC^ mice as these mice do not appear in our former Figure S5.

To increase ease of viewing and ability to distinguish signal from background, we have provided higher magnification insets of areas of interest for these images, consequently these insets also increase appreciation of the lack of *Ifit1* staining in mice lacking *Ifnlr1* (*Ifnlr1*^KO^/*Ifnar1*^Het^). We have also added double-knockout (*Ifnlr1*^KO^/*Ifnar1*^KO^) images to emphasize our conclusions. Finally, we have added quantification of the area of *Ifit1* expression in tissue sections of *Ifnlr1*^Het^/*Ifnar1*^Het^, *Ifnlr1*^KO^/*Ifnar1*^Het^, *Ifnlr1*^Het^/*Ifnar1*^KO^ , and *Ifnlr1*^KO^/*Ifnar1*^KO^ mice.

– Figure 7, crypt cells are not a cell type. There are stem cells, TA and other progenitors present that make up the crypt. This should be renamed to correctly indicate the type of cells in this population.

We have updated our language in the Results section of Figure 7 to more accurately convey the cell types present in the “crypt-associated” grouping that we defined. In certain cases, we refer to a group of cells from the single-cell sequencing data produced by Elmentaite et al. that were annotated as “crypt”. In the interest of accuracy and readability, where applicable in the results of Figure 7, we have clarified the decision to retain the cellular annotations bestowed by the original publishing group and removed any reference to this crypt group as a type of cell.

– Additionally, the scRNA-Seq used for human analysis was a bit strange. This paper used mainly embryos and children to derive the cell types. The fact that only a few cell types were highly present seems to indicate that maybe this was not the best paper. It would be good to compare to stronger papers such as Wang et al. JEM 2020, Fuji et al. Cell Stem Cell 2018. There is also a paper using human ileum showing that stem cells and enterocytes each have their own basal ISGs which are stem cell specific or enterocytes specific (Triana, 2020). The statements about the human data should be toned down or a more robust scRNA data set should be used.

We have not used the data gathered from embryos and have only analyzed the data from healthy children. Rather than add more datasets to the manuscript, we have chosen to tone down our conclusions regarding our analysis of the human scRNA-sequencing data by Elmentaite et al. (Lines 480-482): “Our analysis of a human scRNA-seq dataset is consistent with observations in the murine model, though future studies will be required to definitively address the existence of homeostatic ISGs in human tissue.”

– Figure S6D, why are there different numbers of mice used for each ISG? If the RNA is harvested, then the same RNA can be used to assess all three ISGs. It seems strange that of 30 mice tested for ifit1, only 17 were tested for stat1. These missing numbers seems strange and should be included.

We thank the reviewer for noting the values in Figure S9D (formerly S6D). Upon bringing it to our attention, we realized that the table in question had transposed columns. We have updated the table to display the correct values – for FT rescue, we have tested 30 mice for *Ifit1*, and have tested slightly fewer mice (27 and 24) for *Stat1* and *Oas1a*, respectively.

– The homeostatic ISGs are only found in the tip of the villi, and rotavirus infection normally occurs at the tips of the villi, the authors claim that these homeostatic ISGs are more protective against infection. To make this claim the authors need to perform a co-staining of rotavirus infected tissue to show that the villi that are infected are not the villi which express the homeostatic ISGs. In general, rotavirus staining is needed as the FACS data suggests that a very low numbers of cells are infected which makes it hard to justify the authors claims.

We have now included multicolor in situ hybridization depicting the distribution of *Ifit1* and murine rotavirus at 24hr postinfection (Figure 9I) and have included description of these results in-text (Lines 598-602): “To further contextualize the role of homeostatic ISGs in protection against mRV, we performed in situ hybridization for both mRV and *Ifit1* in the ileum of WT mice at 24 hours post-inoculation (Figure 9I). We observed variability in the magnitude of mRV infection between mice at this timepoint; however, we generally found that mRV RNA was not present in villi that have *Ifit1* expression.”

– The authors claim that IFN is being produced from CD45+ epithelium derived lymphocytes. To support and strengthen this claim, it would be important to perform RNA scope with markers for these cells to determine if the sparse ISG induction seen in the tissue is correlates with the localization of these cells.

Our data showing that IFN-λ is being produced by CD45+ cells associated with the intestinal epithelium is consistent with previous observations by Mahlakoiv et al. (Mahlakoiv 2015, Figure S3). We agree that further defining the cellular source of IFN-λ in response to microbial stimuli is a priority. However, we believe that this question demands robust experimental verification beyond that of correlative imaging and aim to pursue these studies in the future.

– Since the authors claim is that the homeostatic ISGs come from sensing of bacterial components, and that LPS can rescue this phenotype, then the authors should determine if tlr4-/- mice also lack the presence of homeostatic ISGs.

As suggested by the reviewer below, we now include additional data with flagellin stimulation, which is capable of partially restoring homeostatic ISGs. Therefore, we would not expect deficiency of a single TLR to prevent expression of all homeostatic ISGs. In fact, MyD88 deficiency does not uniformly prevent expression of homeostatic ISGs across tissue sites (ileum and colon). Separate studies beyond the scope of the current manuscript will be needed to further fully define the cell types and signaling pathways responsible for homeostatic IFN-λ production.

– Furthermore, the authors claim in the discussion that microbiota components are responsible for the homeostatic levels and that they signal through several MAMPs. However, this is not shown in the paper. In the manuscript the authors only tested LPS. They also used CpG which will signal through MyD88 but does not induce this response. To make the claim that multiple MAMPs can be used, then several other MAMP agonists should be used to show that this is not only a LPS phenotype (e.g. flagellin).

We now include evidence that peroral flagellin is able to stimulate the homeostatic IFN-λ response to a similar extent as LPS in antibiotic-treated mice (Figure S9, formerly Figure S6). We have now included additional commentary regarding this finding in our results (Lines 526-535): “To corroborate these findings, we performed orthogonal analyses of *Ifit1*, *Stat1*, and *Oas1a* expression in ileum tissue of WT mice treated with ABX followed by fecal transplant, LPS administration, administration of the TLR5 agonist: flagellin, or administration of TLR9 agonist: CpG DNA (Figure S9). These data exhibited high variance, but were consistent with imaging data in Figure 8, indicating a partial restoration of homeostatic ISGs in 2050% of ABX-treated mice by LPS, flagellin, or fecal transplant, but not CpG DNA (Figure S9). Together, these data indicate that both LPS and flagellin are sufficient to restore homeostatic ISG expression to a similar extent as total microbiota. The ability of multiple TLR stimuli to partially restore homeostatic ISGs suggests that exposure to a variety of bacterial MAMPs is the basis for localized, homeostatic ISG expression.”

[Editors’ note: what follows is the authors’ response to the second round of review.]

Essential revisions:Functional protection against rotavirusThe reviewers noted the need for quantification of the colocalization of murine rotavirus and Ifit1 expression in our co-staining images. Reviewer 3 also requested additional experimental replicate of Ifit1 imaging at the 96hr post-inoculation timepoint in Figure 9 —figure supplement 2 (previously Figure 9 —figure supplement 1), and MRV colocalization imaging of infection at 96hr post-inoculation.

To address these concerns, we have taken the following actions:

1) We performed an additional experimental repeat of 96hr post-inoculation data in WT mice and included these data in Figure 9 and Figure 9 —figure supplement 2.

2) We now provide quantification of murine rotavirus and *Ifit1* from images (12, 24, and 96hr p.i.) in Figure 9 and Figure 9 —figure supplement 2.

3) Though not specifically requested by reviewers, we also added genome qPCR and quantification of %mRVpositive cells by flow cytometry for the 96hr post-inoculation dataset to increase uniformity in the data presentation across timepoints (Figure 9 —figure supplement 3).

The reviewers noted concerns regarding our interpretation that pre-existing microbiota-dependent Ifnlr1dependent signaling is responsible for protection against murine rotavirus infection.

Our understanding from reading the specific comments of reviewers is that quantitation of co-localization in our images, above, may address their concerns. However, we have also revised the Results section to include quantification of *Ifit1* and mRV co-incidence on lines 665-668; “Quantification of co-staining for *Ifit1* in mRVinfected cells indicated that a minority (~30%) were *Ifit1*+ at 12hr or 24hr post-inoculation, whereas a majority (~63%) of mRV-infected cells were *Ifit1*+ at 96hr post-inoculation (Figure 9I-J and Figure 9 —figure supplement 2).”

Ultimately, we are sympathetic to the reviewers’ puzzlement about how localized, ISG-expressing IECs could provide meaningful protection against viral infection and understand that the reviewers are unconvinced by our discussion of our model in the paper.

We have now addressed alternative models that explain our data in the Discussion section on lines 830844.

“Although we found that homeostatic ISGs provide protection during initiation of mRV infection, it remains unclear how these localized ISG pockets impart this protection. Given our findings in Figure 8 and Figure 8 —figure supplement 1, we suggest that these localized ISGs may be indicative of locations that are particularly vulnerable to viral infection if ISGs were not present at the time of viral exposure. However, alternative explanations for the protective effects that we observe are also plausible, such as: (i) an inability to detect the full magnitude of ISG expression by imaging, or (ii) an unknown temporal component to homeostatic ISG signaling that may be coincident with durable ISG protein expression. Given the magnitude of signal amplification in RNAscope in situ hybridization and the similarity of *Ifit1* expression in our imaging data to *Ifit1* expression in public singlecell sequencing datasets, we find it unlikely that there is more widespread homeostatic ISG expression below the limit of our detection by imaging. However, we do think an uncharacterized temporal component of homeostatic ISG signaling is plausible, wherein individual villi may be rapidly and transiently expressing homeostatic ISGs in response to sensing of bacterial microbiota. This temporal model may also include a more durable ISG protein response that is not fully concordant with expression of ISG transcripts.“

Partial effect of LPS and FlagellinReviewers voiced concerns regarding the interpretation of LPS, flagellin, and fecal-transfer experiments to assess ISG restoration in ABX-treated mice. The primary concern with these data was the validity of interpretation and no experiments were suggested by reviewers.

We have substantially updated our interpretation of data from Figure 8 and Figure 8 —figure supplement 1 in the abstract, results, and discussion by softening conclusions, replacing interpretation with statements of fact in the results, and clarifying our interpretation in the discussion.

Examples include:

i) In the abstract:

Updated Text Lines 28-30: “Furthermore, we assessed the ability of orally-administered bacterial components to restore localized ISGs in mice lacking bacterial microbiota.”

ii) In the figure and section headings of Figure 8 and Figure 8 —figure supplement 1:

Updated Text: Assessing the effect of peroral bacterial products on homeostatic ISGs in ABX-treated mice”.

iii) In the results on Lines 552-554:

Updated Text: “However, localized *Ifit1* expression was visible in 4/12 ABX-treated *Ifnlr1*^flox/flox^ mice administered LPS (Figure 8E, 8H-I).” (iv) In the discussion for example on lines 786-789:

Updated Text: “Given the results of our data in Figure 8 and Figure 8 —figure supplement 1, we suggest that LPS administration, flagellin administration, or fecal transplant can partially restore the expression of homeostatic ISGs in the small intestine. This interpretation supports the concept…”.